# Strategies to enable large-scale proteomics for reproducible research

Rebecca C. Poulos [1,15], Peter G. Hains [1,15], Rohan Shah [1,15], Natasha Lucas[1], Dylan Xavier [1], Srikanth S. Manda [1], Asim Anees[1], Jennifer M. S. Koh[1], Sadia Mahboob [1], Max Wittman[1], Steven G. Williams [1], Erin K. Sykes [1], Michael Hecker[1], Michael Dausmann [1], Merridee A. Wouters [1], Keith Ashman [2], Jean Yang [3], Peter J. Wild [4,5], Anna deFazio [6,7,8], Rosemary L. Balleine [1], Brett Tully [1], Ruedi Aebersold [9,10], Terence P. Speed [11,12], Yansheng Liu [13,14], Roger R. Reddel [1], Phillip J. Robinson [1] & Qing Zhong [1✉]

Reproducible research is the bedrock of experimental science. To enable the deployment of large-scale proteomics, we assess the reproducibility of mass spectrometry (MS) over time and across instruments and develop computational methods for improving quantitative accuracy. We perform 1560 data independent acquisition (DIA)-MS runs of eight samples containing known proportions of ovarian and prostate cancer tissue and yeast, or control HEK293T cells. Replicates are run on six mass spectrometers operating continuously with varying maintenance schedules over four months, interspersed with ~5000 other runs. We utilise negative controls and replicates to remove unwanted variation and enhance biological signal, outperforming existing methods. We also design a method for reducing missing values. Integrating these computational modules into a pipeline (ProNorM), we mitigate variation among instruments over time and accurately predict tissue proportions. We demonstrate how to improve the quantitative analysis of large-scale DIA-MS data, providing a pathway toward clinical proteomics.

[1] ProCan®, Children's Medical Research Institute, Faculty of Medicine and Health, The University of Sydney, Westmead, NSW, Australia. [2] Sciex, 2 Gilda Court, Mulgrave, VIC, Australia. [3] School of Mathematics and Statistics, The University of Sydney, Sydney, Australia. [4] Dr. Senckenberg Institute of Pathology, University Hospital Frankfurt, Frankfurt am Main, Germany. [5] Department of Pathology and Molecular Pathology, University Hospital Zurich, Zurich, Switzerland. [6] Centre for Cancer Research, Westmead Institute for Medical Research, Westmead, NSW, Australia. [7] Faculty of Medicine and Health, The University of Sydney, Westmead, NSW, Australia. [8] Department of Gynaecological Oncology, Westmead Hospital, Westmead, NSW, Australia. [9] Department of Biology, Institute of Molecular Systems Biology, ETH Zürich, Zürich, Switzerland. [10] Faculty of Science, University of Zürich, Zürich, Switzerland. [11] Bioinformatics Division, Walter and Eliza Hall Institute of Medical Research, Parkville, VIC, Australia. [12] Department of Mathematics and Statistics, University of Melbourne, Melbourne, VIC, Australia. [13] Department of Pharmacology, Yale University School of Medicine, New Haven, CT, USA. [14] Yale Cancer Biology Institute, Yale University, West Haven, CT, USA. [15] These authors contributed equally: Rebecca C. Poulos, Peter G. Hains, Rohan Shah. ✉email: qzhong@cmri.org.au

Precision medicine relies on the ability to distinguish patients according to their likely clinical outcome. In oncology for example, the detection of specific features of patho-biology in cancer tissues has become key, as targeted therapies are increasingly used in clinical trials and practice. There are established tissue biomarkers in routine use, such as estrogen receptors or *HER2* in breast cancer[1], that determine management plans for individual cancer patients. However, clinically useful predictive biomarkers are unknown for the majority of conventional or targeted anti-cancer agents and this remains an area of unmet clinical need.

Studies of cancer and other tissues by multi-dimensional omic—in particular genomic and transcriptomic—analyses, have greatly increased our understanding of the molecular basis of cancer and other diseases. Until very recently, it has not been technically feasible to match sample-level genomic data with proteomic analysis. However, the field of tissue proteomics is currently in a phase of rapid evolution, based primarily on advances in mass spectrometry (MS) technology[2,3]. MS techniques that are applicable to biopsy-sized tissue samples have now been established and the availability of tissue proteomics, in combination with other 'omic data, holds great promise for biomarker discovery[4–9]. To build the comprehensive proteomic knowledgebases that are required to underpin the application of this technology to clinical medicine, major technical and analytical challenges need to be addressed[4].

Data-independent acquisition (DIA)[10,11] methods such as sequential windowed acquisition of all theoretical fragment ion spectra (SWATH™)-MS are ideally suited to high-throughput tissue research because they result in comprehensive peptide data capture, and hence a permanent digital map of the proteome that can be re-interrogated over time[2,12–14]. Currently, DIA-MS datasets can be analysed by various proteomic software pipelines with high concordance, and false discovery rates (FDR) are used to control error propagation[15,16]. When measurements are acquired over short experimental periods, DIA-MS data collected across different laboratories can be adequately combined[15,17]. However, to achieve datasets of sufficient size to support robust discovery, data collected from multiple instruments over long periods must be effectively integrated. It is therefore imperative that the impact of factors affecting the reproducibility of peptide quantitation are known, and that data analysis techniques are developed to optimise data integration over time and across instruments.

To enable acquisition of reproducible large-scale data for studies of cancer and other diseases, the aim of this study is first to document the degree of long-term inter-instrument and temporal variation in the discriminative proteomic profiles of cancer tissues analysed in a high-throughput facility. We then seek to develop methods to improve reproducibility by enabling the integration of temporally disparate data. To do so, data are acquired from a single series of standardised samples that are collected over 1560 DIA-MS runs on six Quadrupole Time-Of-Flight (QTOF) mass spectrometers operating in a single laboratory over a four-month period, during which ~5000 other samples are also run. In the absence of significant sample variability, patterns of system-level variation are then used to develop methods for data normalization and missing value replacement that substantially improve predictive accuracy.

## Results

**Study design and data acquisition.** Analysing DIA-MS measurements made across instruments, over extended periods of time, and in large cohorts with variable intensities, is a major challenge. To address this, an experiment was designed to assess the reproducibility of DIA-MS measurements collected from different mass spectrometers in a single facility over a period spanning approximately four months. The design enables the analysis of technical variation in the absence of sample variation in a large dataset. Eight experimental samples were prepared: Samples 1–6 were a dilution series of ovarian cancer tissue (0%, 3.125%, 6.25%, 12.5%, 25% and 50%) offset by yeast and a fixed proportion (50%) of prostate cancer tissue; Sample 7 was a 1:1 mix of ovarian cancer tissue and yeast cells; Sample 8 was a human cell line (HEK293T) used as control (Fig. 1a). Samples were run either in triplicate (the core of the dilution series) or in duplicate, in a defined sequence comprising sets of 20 samples (Fig. 1b). DIA-MS data were acquired with 90-minute gradient lengths at the Australian Cancer Research Foundation International Centre for the Proteome of Human Cancer (ProCan®)[4] on six SCIEX™ TripleTOF® 6600 QTOF mass spectrometers. The 20-sample set was run on each instrument 13 times at spaced intervals over a four-month period (Fig. 1c), where each data collection occurred over 48 h. At study commencement, the 20-sample set was run four times in succession (days 1, 3, 5 and 7), then once per week for the remainder of the month (commencing on days 14, 21 and 28), and then once per month for the remainder of the first three months (commencing on days 56 and 84) (Fig. 1c). After each instrument underwent a major clean, sample sets were run a further four times in succession (commencing on days 101, 103, 105 and 107) (Fig. 1c). In total 1560 DIA-MS runs were performed. To mimic a real-world scenario relating to data acquisition, mass spectrometer maintenance schedules varied according to each individual instrument's performance (Supplementary Fig. 1a, b). Actual days of instrument maintenance and a few minor deviations from experimental design are shown in Supplementary Fig. 1b. To reflect steady-state operation of a high-throughput facility, data collection for this study was interspersed with ~5000 runs of unrelated samples.

A spectral library was generated by merging search results from three search engines (Mascot[18], X!Tandem[19] and MSGF+[20]) using PeptideShaker[21] and Skyline[22] (see Methods). The MS raw data files were then processed with reference to this spectral library using OpenSWATH[23] and PyProphet[24] with 1% FDR at both global peptide and protein levels for identification and 5% run-specific peak group FDR for quantification (Supplementary Fig. 1c; see Methods). After removing files (*n* = 26) that did not pass baseline quality controls such as total numbers of proteins identified and correlation with replicate samples (see Methods), and discarding a small number of peptides that were inconsistently observed across the cohort (*n* = 934; see Methods), 1,527 runs were retained (Supplementary Data 1, 2). These runs comprised data from a total of 17,054 peptides derived from 2796 proteins, of which 2363 were human proteins. All proteins were identified by unique peptides (an average of 6.1 peptides were identified per protein), and 80% of proteins were supported by evidence from at least two peptides (Supplementary Fig. 2a). The experimental FDR per sample approximated 1.5% (Supplementary Fig. 2b). Many peptides were consistently identified in technical replicates (Supplementary Fig. 2c) and peptide intensities were broadly consistent across the six instruments and the eight samples (Supplementary Fig. 2d, e). Of 13,485 likely true-positive peptides (i.e., peptides observed in >10% of *n* = 151 replicates of Sample 8, derived from HEK293T cells), *n* = 10,109 peptides (75%) were observed in at least 75% of the replicates.

**Baseline data reproducibility during a single week.** We first investigated the reproducibility and accuracy of data collected over a short timeframe by analysing the final four sample set runs of the study, occurring on each MS after instrument cleaning. Principal component analysis[25] revealed separation of samples

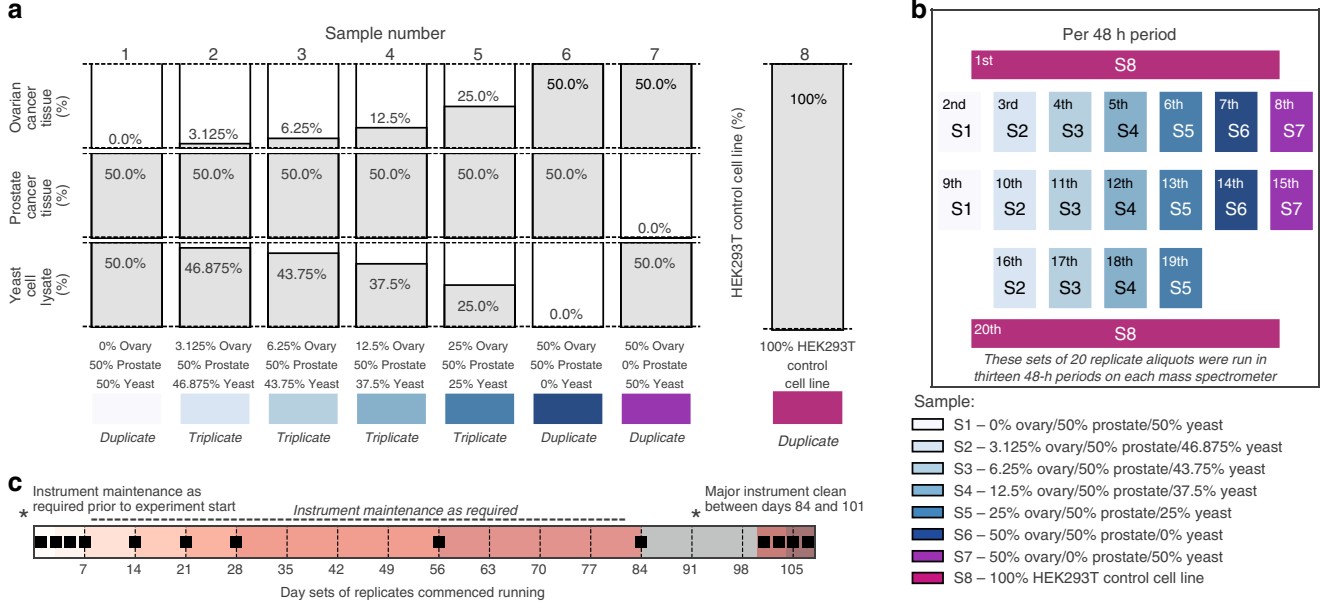

**Fig. 1 Study design. a** Composition of the eight samples analysed repeatedly throughout the study. **b** Twenty mass spectrometry runs during thirteen 48-h periods on each of six instruments. Each run is represented by a coloured panel corresponding to each of the eight samples (labelled S1–S8), with the run order indicated in the upper left corner. Four samples were run in duplicate and four in triplicate during each 48-h period. **c** Mass spectrometer scheduling. Days on which 48-h periods of data collection commenced are indicated with a black bar, and the intended instrument cleaning schedule is also indicated.

and negligible instrument batch effects (Fig. 2a). Without data normalisation, the median coefficient of variation (CV) of peptides detected in more than 80% of MS runs from each of Samples 1–7 ($n = 2,950$ frequently observed peptides; see Methods) was ≤20% per instrument for replicates of each sample (Fig. 2b, Supplementary Fig. 2f). We observed an approximately linear relationship between the intensities of ovarian cancer tissue-specific peptides (see Methods) and peptides from yeast proteins according to the known relative proportions of ovarian cancer tissue and yeast, respectively (Fig. 2c). Median fold changes in the intensity of peptides from yeast proteins approximated expected differences (Supplementary Fig. 2g). To investigate consistent detection of the basal tissue matrix, we examined prostate-specific antigen (PSA; the protein used in prostate cancer screening[26]), and found similar intensities among PSA peptides in Samples 1–6 containing 50% prostate cancer tissue, along with some likely false positive observations in Sample 7 containing no prostate tissue (Fig. 2d). Finally, we observed an association between the intensity of peptides from a housekeeping protein and the known proportion of human tissue in each sample (TAR DNA-binding protein 43[27]; Fig. 2e). From these initial qualitative analyses, we confirm that DIA-MS data obtained over a short time-period are generally reproducible and can support discriminative accuracy between human tissues prior to normalisation, as indicated by previous studies[17].

**Decrease in instrument sensitivity over time since cleaning**. We next describe the longitudinal variation in experimental data over the entire study period, as was the primary aim of this study. Median peptide intensities varied considerably during the second and third experimental months, consistent with the anticipated degradation in instrument sensitivity related to increased time after cleaning (Fig. 3a and Supplementary Fig. 3a). To demonstrate the effects of long experimental periods on peptide intensities, a single human peptide that appeared to be ovarian cancer tissue-specific was examined in detail. Considering only data acquired on a single instrument during a single day, the intensity

of this peptide exhibited a strong Pearson correlation ($r$) with ovarian cancer tissue proportion ($r ≥ 0.98$; Fig. 3b). The strength of this correlation was reduced when measurements were combined across the experimental period on a single instrument ($r = 0.87$) and further across all instruments ($r = 0.84$) (Fig. 3b). This example demonstrates how variation over time and between instruments can greatly compromise quantitative reproducibility and could limit the ability to discriminate between samples. In the next sections, we describe two computational strategies to overcome barriers relating to data normalisation and missing values.

**Development of a method for data normalisation**. To minimise the impact of technical variation on peptide intensities over time and across instruments, we developed a normalisation strategy Remove Unwanted Variation III Complete (RUV-III-C) to remove unwanted variation from complete (i.e., non-missing) intensities of each peptide in a data matrix containing missing values. RUV-III-C is thus a variant of RUV-III[28]—an existing normalisation approach that only allows input of a data matrix without missing values. RUV-III-C relies on technical replicates and negative control variables to estimate and then subtract unwanted variation from a dataset.

To simulate a common experimental scenario, we randomly assigned groups of triplicates acquired on different instruments as technical replicates for the purposes of RUV-III-C (see Methods). As negative controls, we selected peptides from yeast proteins ($n = 1622$) and scaled these such that their mean intensities were identical across samples. In this manner, we only normalised samples containing yeast (Samples 1–5 and 7). Applying RUV-III-C to our dataset (see Methods), we found that longitudinal peptide intensity variations were largely removed (Fig. 3c and Supplementary Fig. 3b). Despite the extended period for data collection, variation of the single peptide across the entire experiment was greatly reduced ($r ≥ 0.98$; Fig. 3d).

It is possible for normalisation to obscure biological changes when removing technical variation. This is a particular risk when cohorts comprise complex samples such as heterogeneous human

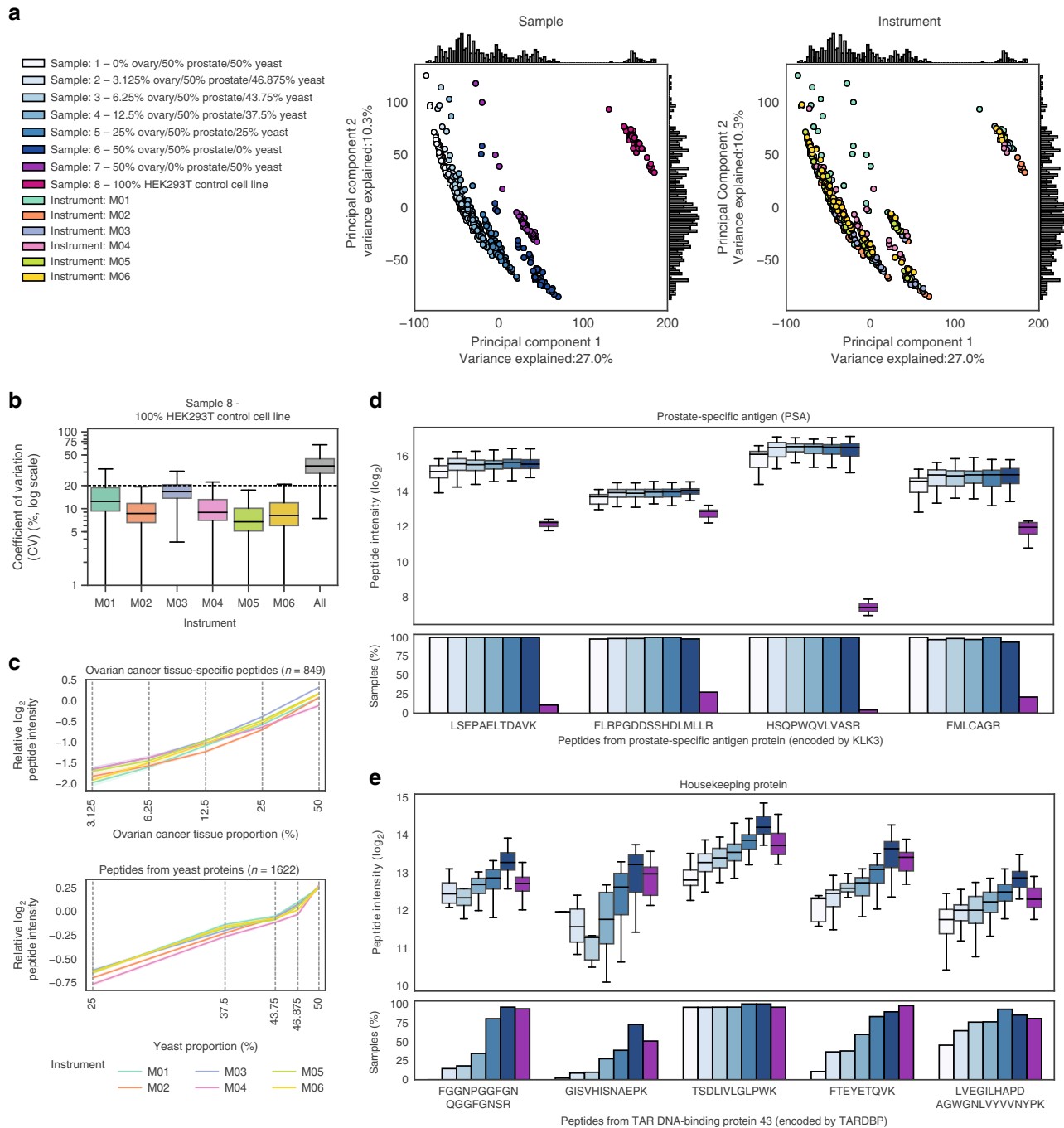

**Fig. 2 Baseline DIA-MS data reproducibility.** Data shown in all plots were acquired during the experimental week after instrument cleaning (days 101, 103, 105 and 107) and were not normalised. **a** Principal component analysis of $log_2$-transformed experimental data, with data points coloured by sample (left) and instrument (right). Missing values were filled with zeros. **b** Coefficient of variation (CV) per instrument in the HEK293T control cell line (Sample 8). CV was calculated using frequently observed peptides ($n = 2950$ peptides). A black dashed line marks a CV of 20% for reference. **c** Relative $log_2$-transformed intensities per sample of ovarian cancer-tissue specific peptides (upper) and peptides from yeast proteins (lower), coloured by instrument. The mean peptide intensity from each sample was adjusted so that relative intensities are comparable, by dividing each value by the overall mean peptide intensity measured on a given instrument during the period. Ovarian cancer tissue and yeast proportions are plotted on the $log_2$-scale. **d**, **e** Intensities of all peptides identified from **d** the prostate-specific antigen (PSA) protein encoded by *KLK3* and **e** the housekeeping protein encoded by *TARDBP*. Boxplots show peptide intensity, with bar plots indicating the proportion of replicate samples in which each peptide was observed. Plots are coloured according to sample, using colour-codes as shown in **a**. For replicate numbers *n*, refer to Supplementary Data 2. In **b**, **d** and **e**, the box indicates quartiles and the whiskers indicate the rest of the distribution, with outliers not shown. Source data are provided as a Source data file.

tissues. To assess RUV-III-C in this respect and to measure the effect across a larger number of peptides, we examined the relationship between the intensity of all frequently observed human peptides with ovarian cancer tissue content. The median

Pearson correlation across each peptide increased from 0.25 before normalisation to 0.62 after normalisation, consistent with an improvement in specific signal detection (Fig. 3e). Median normalisation and median normalisation plus ComBat[29] for

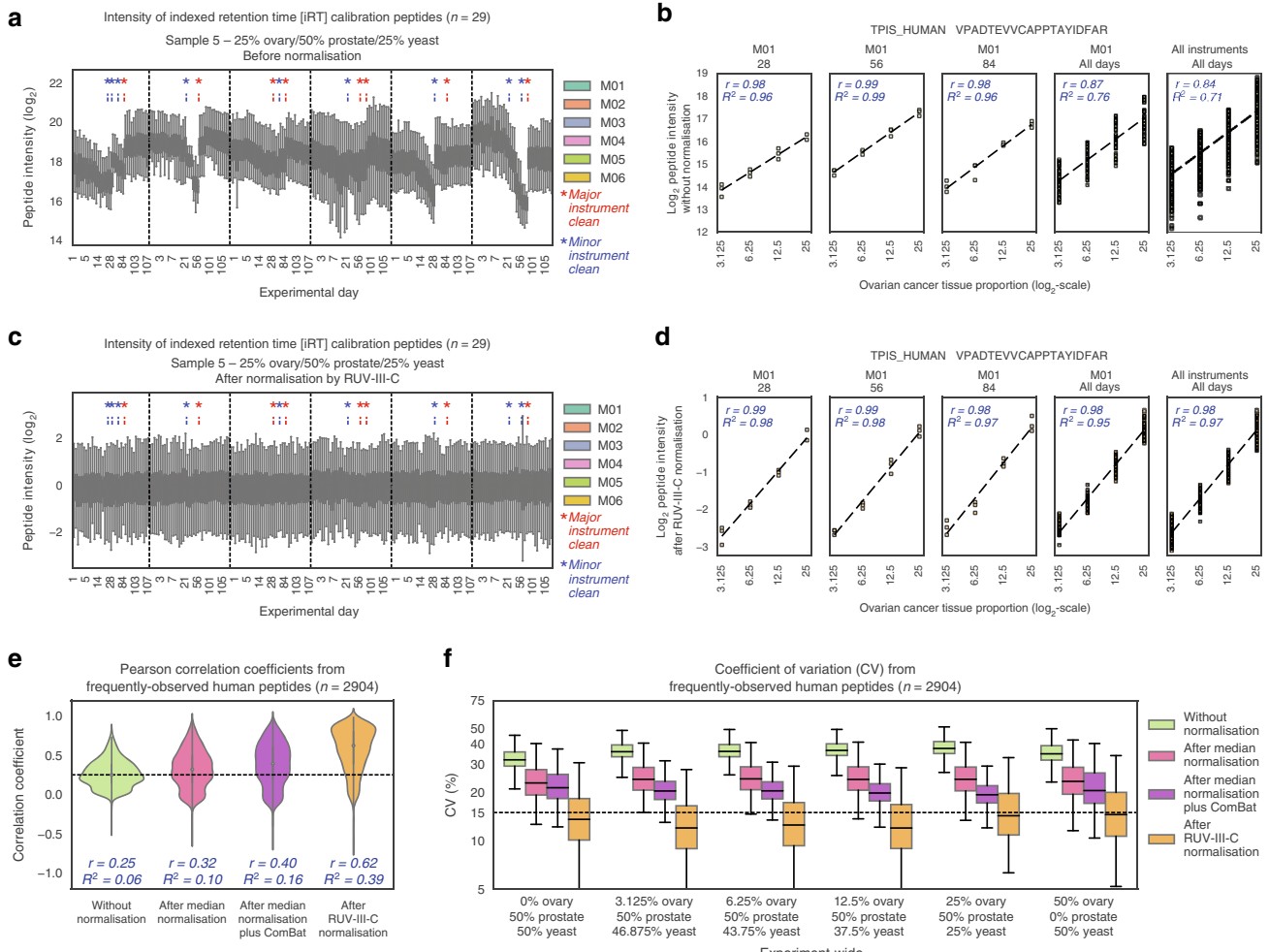

**Fig. 3 Peptide intensity variation during the experimental period and normalisation approaches. a**, **c** Intensities of indexed retention time calibration peptides ($n = 29$) in replicate runs of Sample 5 (containing 25% ovarian cancer tissue/50% prostate cancer tissue/25% yeast), **a** before normalisation and **c** after RUV-III-C normalisation. Boxplots are coloured by instrument, within which data are ordered from earliest experimental day (left) to latest experimental day (right). Maintenance schedules of major (red) and minor (blue) instrument cleaning are indicated. Only every sixth experimental day is labelled on the horizontal axis. **b**, **d** Intensity of a single human peptide **b** before normalisation and **d** after RUV-III-C normalisation. Pearson correlation ($r$) and $R^2$ are shown in italicised blue text, and the black dashed line indicates the predicted association from a linear regression model. **e** Correlation coefficients from Pearson correlation of each frequently observed human peptide ($n = 2904$) with ovarian cancer tissue proportions. Median Pearson correlation ($r$) and $R^2$ from each distribution are shown in italicised blue text. A black dashed line indicates the median correlation coefficient before normalisation. Correlations were calculated using $\log_2$-transformed peptide intensities and ovarian cancer tissue proportions. **f** Coefficient of variation (CV) of frequently observed human peptide intensities, calculated for each sample during the experimental period across all instruments. A black dashed line marks a CV of 15% for reference. In **a**, **c** and **f**, the box indicates quartiles and the whiskers indicate the rest of the distribution, with outliers not shown. Source data are provided as a Source data file.

batch effect removal at first appeared to successfully normalise the entire set of peptides (Supplementary Fig. 3c), but importantly, variability remained in the intensities of individual peptides (Supplementary Fig. 3d). Further, correlation with the human tissue dilution series was only minimally improved (0.32 and 0.40, respectively; Fig. 3e). The experiment-wide median CV of frequently observed human peptides reduced from 37% before normalisation to 24% after median normalisation and 20% with ComBat, while RUV-III-C produced the lowest experiment-wide CV of 13% (Fig. 3f). In summary, RUV-III-C removed unwanted variation introduced over an extended experimental period on multiple instruments while elevating important biological signal.

To demonstrate its utility across independent datasets and MS platforms, we applied RUV-III-C to two previously published datasets. First, we considered a multi-laboratory assessment of DIA-MS in which 30 peptides were diluted into a background of

HEK293 cell lysate[17]. Samples in this study were acquired on SCIEX™ TripleTOF 5600/5600+ instruments. Using this dataset, RUV-III-C produced the lowest median CVs for these peptides (average 0.7% across 11 laboratories worldwide; Supplementary Fig. 3e), which was consistently lower than before normalisation (14%), after median normalisation (9%; this method was used in the original analysis[17]) and RUV-III (3%) (Supplementary Fig. 3e). Correlations with the known human tissue dilution series were also highest with RUV-III-C ($r = 0.89$) when compared against other methods ($r \le 0.81$; Supplementary Fig. 3f). Second, we considered a dataset of 1508 plasma samples[30] acquired on an Orbitrap Fusion Lumos Tribrid mass spectrometer from Thermo Scientific™. RUV-III-C was able to successfully remove batch effects, produce low coefficients of variation and reproduce essential findings from the study (see Supplementary Note 1 and Supplementary Figs. 4 and 5). This demonstrates

the utility of RUV-III-C for normalisation across different DIA-MS instrument platforms.

**Technical replacement of missing values**. Missing values are a challenge in proteomics when combining large datasets. To define a method to address these, we first examined the types of missing values present in our study, and then investigated a method to reduce missing data by effectively utilising technical replicates. Large proteomic datasets can harbour missing values for one of three reasons: (a) true missing (true negative) peptides that are not present in a sample, (b) missing not at random (MNAR) when missing values are associated with peptide intensities (for example, a low-intensity peptide falling below the limit of detection of the instrument), or (c) missing completely at random (MCAR) when features are not detected despite ions being present at detectable concentrations. We first sought to identify MNAR and MCAR peptides in our dataset by leveraging our experimental design comprising multiple technical replicates, reasoning that peptides with many missing values across replicates may have intensities around an instrument's limit of detection (i.e., MNAR). We discarded peptides that may be true missing by removing those that were missing in ≥90% of total replicates from a given sample. We then selected likely MNAR and MCAR peptides as those with missing intensities in an average of ≥5 of 6 and ≤1 of 6 replicates, respectively, across random groups of six replicates (see Methods). The MNAR peptides had mean non-missing intensities that were significantly lower than MCAR peptides ($P < 0.0001$ by unpaired $t$-test; Fig. 4a, Supplementary Fig. 6a), demonstrating that with certain study designs these types of missing values can be distinguished.

The numbers of peptides observed across replicates varied up to three-fold during the study, with highest peptide numbers recorded after instrument cleaning (Fig. 4b). Adequately dealing with missing values in a large dataset can be challenging. We developed a method termed technical replacement that leverages technical replicates at different phases of instrument maintenance, to replace missing values with plausible intensities. For the purposes of technical replacement, we assigned MS runs into random smaller groups of technical replicates. These groups ranged in size from between two to six replicates for each iteration of technical replacement, with replicates always acquired on different instruments (see Methods). We then replaced missing values with a value sampled from a normal distribution centred around the mean normalised intensity observed for that peptide if it were identified in a technical replicate (see Methods). We found that the effects of instrument maintenance cycles were minimised with three technical replicates (when replacement only occurred for peptides observed in two replicates) and were almost entirely removed with six technical replicates (no constraints on replacement) (Supplementary Fig. 6b, c). We defined likely true and false positives (see Methods) and, after technical replacement, the likely true positive rate increased from 97.2% to over 99% (Fig. 4c). With no constraints on replacement, the likely false positive rate increased with each subsequent technical replicate, from 4.8% to 24.6% with six replicates (Fig. 4c). Hence, to constrain false positive accumulation, we conclude that missing values are best replaced only when a peptide is observed in more than one replicate. Only in this manner can the true positive rate be increased without considerably impacting upon the false positive rate. When technical replacement was applied across replicates acquired on three instruments (where missing values were only replaced when a peptide was observed in two replicates), ~20% of missing values could be replaced with plausible non-zero intensities. Technical replacement became more effective as samples comprised increasing proportions of

ovarian cancer tissue (Fig. 4d), likely because higher peptide intensities led to more reliable detection of each peptide in at least two replicates.

Acquiring triplicates on different instruments rather than the same instrument resulted in greater numbers of peptide identifications (Supplementary Fig. 6d). Relevant to laboratories where only one instrument is available for a given study, acquiring technical triplicates randomly throughout the experimental period rather than sequentially was able to more effectively minimise the effects of instrument performance degradation (Supplementary Fig. 6e).

We defined the combined use of RUV-III-C and technical replacement on a proteomic dataset as ProNorM (Proteomics Normalisation and Missingness Removal).

**Simulation of cohort analyses in proteomics**. One goal of clinical proteomics is the detection of statistically significant differences between patient groups. Clinical cohorts and tissue samples are inherently heterogeneous, and technical variation makes differentiation even more challenging. Examining frequently observed human peptides and averaging across technical triplicates, we found that ProNorM conferred a vast improvement on our ability to detect and quantify significant differences in human peptide intensities between samples containing different proportions of human tissue ($P < 0.0001$ by unpaired $t$-test; Fig. 5a). Neither median normalisation nor median normalisation plus ComBat produced improvements of such a scale (Supplementary Fig. 7a). Our findings illustrate how ProNorM can improve the quantitative reproducibility of data acquired over a long time period, enabling the improved detection of biologically meaningful differences between samples.

We next determined the cohort sizes required to overcome technical variability in a DIA-MS dataset after ProNorM. We averaged peptide values obtained from technical triplicates and then, beginning with the smallest proportionate increase from 3.125% (Sample 2) to 6.25% ovarian cancer tissue (Sample 3) in a 50% prostate cancer tissue background, we found that 25 samples enabled the detection of significant differences in an average of >50% of frequently observed human peptides (Fig. 5b). For a larger step increase from 12.5% (Sample 4) to 25% (Sample 5), 25 samples in triplicate enabled the detection of significant differences in an average of >90% of frequently observed human peptides (Supplementary Fig. 7b). Without ProNorM, the sample sizes required to achieve statistical significance were considerably larger (Fig. 5b and Supplementary Fig. 7b).

**Prediction of tissue proportion in a complex mixture**. The ability to estimate the proportion of a specific tissue component in a complex sample could have diagnostic utility (for example, the proportion of cancer cells in a tissue biopsy)[31]. To demonstrate whether such an analysis might be possible using DIA-MS data, we applied machine learning to estimate the proportion of ovarian cancer tissue among the prostate cancer tissue background after ProNorM. We trained four regularized regression models on the entire set of human peptides measured in three concentrations (for example, 6.25%, 12.5% and 25% ovarian cancer tissue; $n = 695$ samples) to avoid overfitting. Hyperparameters were tuned by ten-fold cross validation, and we report out-of-sample errors after testing the models on the remaining, independent concentration (for example, 3.125% ovarian cancer tissue concentration; $n = 225$ samples; see Methods; Fig. 6 and Supplementary Fig. 8). Our models successfully predicted ovarian cancer tissue proportions with a mean absolute error of as low as 0.8% (Fig. 6). Therefore, with appropriate training data, DIA-MS could be applied to measure tissue proportions in a complex

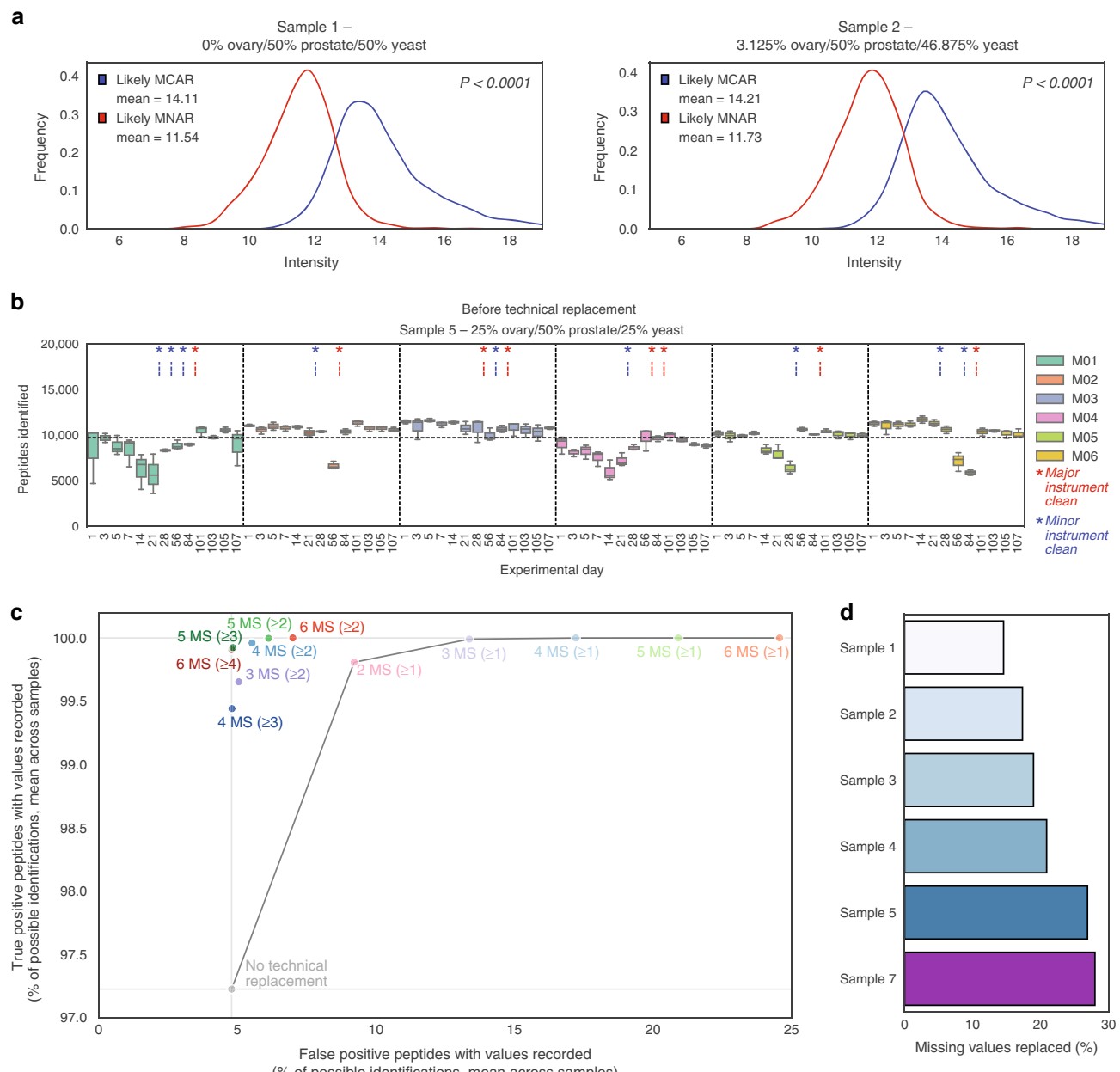

**Fig. 4 Missing values and results from technical replacement. a** Distribution of median non-missing intensity of each peptide designated as likely missing completely at random (MCAR) and missing not at random (MNAR) in Samples 1 and 2. *P*-value determined by two-sided unpaired *t*-test. **b** Peptides identified in each replicate per experimental day in replicate runs of Sample 5 (containing 25% ovarian cancer tissue/50% prostate cancer tissue/25% yeast). Boxplots are coloured by instrument, within which data are ordered from earliest experimental day (left) to latest experimental day (right). Maintenance schedules of major (red) and minor (blue) instrument cleaning are indicated by asterisks. The box indicates quartiles and the whiskers indicate the rest of the distribution, with outliers not shown. A horizontal dashed line indicates the mean number of identifications across the experimental period. For replicate numbers n, refer to Supplementary Data 2. **c** Mean percentage of possible true and false positive peptide identifications across samples after each method of technical replacement. Each method is denoted by first indicating the number of instruments (MS) and then the number of replicates in which a peptide must have been observed for technical replacement to occur (bracketed). **d** Proportion of missing values replaced in each sample after technical replacement. Data are shown for triplicates, with missing values replaced when a peptide was observed in two of three replicates, i.e., 3 MS (≥2 MS). Source data are provided as a Source data file.

background, demonstrating significant potential for clinical proteomics.

## Discussion

This study of data collected from 1560 DIA-MS runs of complex samples, in a single facility over four months, provided an opportunity to develop approaches that enable large-scale tissue proteomic data to be mined for improved reproducibility over

time and between instruments. Tissue samples were selected to ensure that a sufficient amount of peptide was available to complete the large number of MS runs, and the experiment was not optimised for an in-depth analysis of the prostate or ovarian cancer proteome.

Consistent with a multi-laboratory study demonstrating a high level of analytical concordance between different locations analyzing a reference sample over a single week[17], we found data

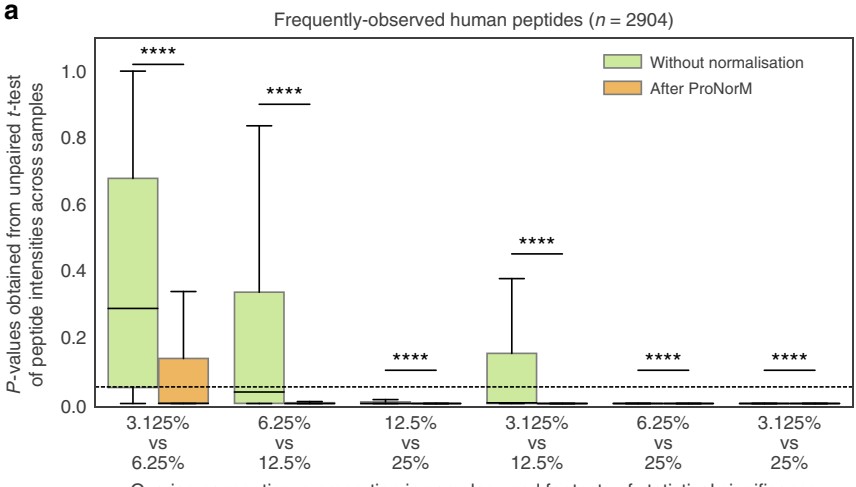

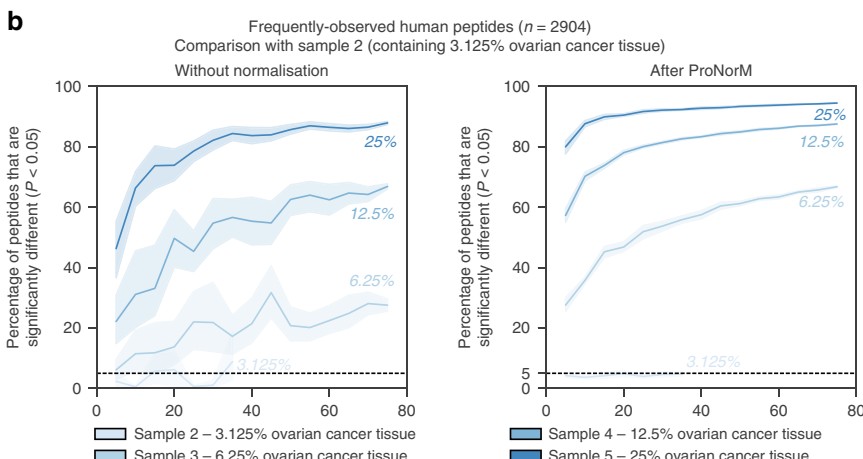

**Fig. 5 Simulation of cohort analyses for discovery proteomics. a** Distribution of *p*-values obtained from unpaired two-sided *t*-test of intensities of frequently observed human peptides (*n* = 2904). Boxplots show *p*-values obtained when comparing samples containing the ovarian cancer tissue proportions indicated (ranging from 3.125% to 25%). Results are shown before normalisation (green) and after ProNorM (orange). The box indicates quartiles and the whiskers indicate the rest of the distribution, with outliers not shown. A black dashed line indicates a *p*-value of 0.05 and **** denotes *P* < 0.0001. **b** Percentage of frequently observed human peptides that were significantly different (vertical axis) in simulated cohorts of varying sizes (horizontal axis). Plots show comparison between Sample 2 (containing 3.125% ovarian cancer tissue) and Samples 2–5 (containing 3.125–25% ovarian cancer tissue), without normalisation (left) and after ProNorM (right). Shading denotes 95% confidence intervals derived from ten iterations of random selections of replicates of each sample. For statistical tests in both (**a**, **b**), the mean of each peptide was first calculated within each set of assigned technical triplicates. Source data are provided as a Source data file.

collected intensively from one or more instruments over a relatively short time frame to be reproducible and able to discriminate between samples with even minor relative differences. By also acquiring data over an extended period of time however, our study demonstrated substantial deterioration in MS measurement sensitivity as instruments approached the threshold for a maintenance intervention. The impact of this phenomenon on data reproducibility was two-fold: the coefficient of variation in replicate measurements was increased to the point where discrimination between disparate samples may be obscured, and the number of quantified peptides was reduced as more peptides fell below the limit of detection. Most importantly, our unique study design enabled insights into the sources of technical variation in high-throughput MS, leading to the development of the ProNorM methodology that can mitigate these effects particularly over long periods and/or across multiple instruments.

Proteomic data inevitably contain missing values, with these arising even when data are obtained by state-of-the-art DIA acquisition and processing pipelines because of limitations imposed by technical noise, inaccurate peak-picking algorithms and incorrect computation of false discovery rates[32–35]. Deficiencies are exacerbated by the effects of mass spectrometer or liquid chromatography column maintenance, signal-to-noise ratios, mass calibration calculations and retention time offsets. Missing values pose a challenge for most forms of analysis of proteomic data including normalization. ProNorM involves a normalization method that can correct for technical variation without undue influence from missing values. The effectiveness of RUV-III-C in normalizing replicate data was assessed with a variety of metrics, such as median experiment-wide CV of frequently observed human peptides which decreased from 37% to 13%, and dilution linearity which improved from 0.25 to 0.62 Pearson correlation (see also Fig. 5 and Supplementary Fig. 7).

ProNorM utilises a technical replacement strategy that leverages peptide detection in technical replicates to replace ~20% of missing values with plausible non-zero intensities. A caveat of this approach is that a minimum of three technical replicates are desirable. Increasing technical replicate MS runs clearly comes at

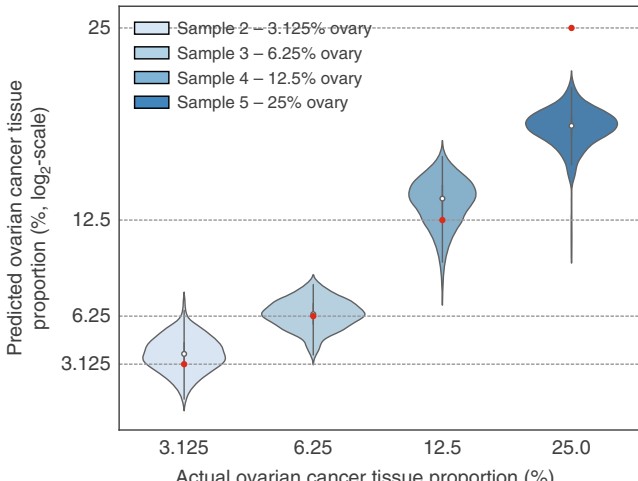

**Fig. 6 Proportion of ovarian cancer tissue predicted by a multilayer perceptron regressor.** Violin plots indicate the ovarian cancer tissue proportions predicted by a multilayer perceptron regressor model. The expected ovarian cancer tissue proportion for each sample is marked by a red data point. Source data are provided as a Source Data file.

a cost. Increasing the number of technical replicates may be a useful strategy for analysis of small clinical cohorts, such as those available from rare diseases or clinical trials, or where researchers would derive benefit from a data matrix with fewer missing values. In the absence of technical triplicates for all samples, RUV-III-C can still be applied to a cohort independent of technical replacement, provided that at least some samples are replicated at various points during the experimental period.

Our results have other practical implications for the design of large-scale studies of complex samples such as cancer tissues. Our experimental design was not intended to focus on assessing the reproducibility of sample preparation. Nor did the design enable the effects of MS and liquid chromatography performance to be fully distinguished, and performance benefits may be achieved by varying the existing maintenance schedules of each of these pieces of equipment. We show that technical replicates are best run on multiple instruments at different stages of performance degradation. Where multiple instruments are not available, running technical replicates at disparate times rather than consecutively is beneficial. This has the dual benefit of maximising peptide identifications for technical replacement and ensuring that replicates measure the greatest amount of unwanted variation for removal. Since the only requirement for technical replacement is the existence of technical replicates, the replacement strategy is not limited to proteomics but could be applicable to other quantitative measurements that are prone to missing values.

In summary, we show how large-scale and longitudinally collected DIA-MS data can be processed to enable substantial improvements in reproducibility. We demonstrate that our results are applicable to MS measurements acquired on different DIA-MS instrument platforms. Our study guides the design of future large-scale MS experiments, demonstrating that multiple instruments can be effectively utilised in the same study, and illustrating gains from improved experimental design and computational approaches. Moreover, ProNorM enables the detection and quantification of small biological differences between samples and, with appropriate training data, the detection of as little as ~3% of one tissue within another. This rich dataset can be mined again by other researchers to benchmark future computational endeavours and methodological advances. Our findings will become increasingly important as the scale of clinical studies

expands, with technological advances allowing shorter gradient times and even smaller sample quantities. This study provides a pathway toward future precision medicine applications by enabling quantitatively accurate and comparable proteomic results from patient cohorts.

## Methods

**Preparation of human cancer tissue samples**. Ovarian serous carcinoma tissue was obtained from the Gynaecological Oncology Biobank (GynBiobank), Western Sydney Local Health District at Westmead. GynBiobank recruits unselected, consecutive clinic-based donors being investigated for, or diagnosed with gynaecologic cancer. Most biospecimens are collected at primary surgery. Participants provide written informed consent for the use of biospecimens in research [HREC92/10/4.13]. Prostate cancer tissue was accessed from the Department of Pathology and Molecular Pathology at the University Hospital Zurich, Switzerland. Men with clinically localised prostate cancer who were scheduled for radical prostatectomy were selected from a cohort of 1200 patients within the single-centre Prostate Cancer Outcomes Cohort (ProCOC) study[36,37]. This study was approved by the Cantonal Ethics Committee of Zurich, the associated methods were carried out in accordance with the approved guidelines, and each patient signed an informed consent form [KEK-ZH-No. 2008-0040]. The prostate cancer tissue was embedded in optimal cutting temperature (OCT) compound. The use of ovarian and prostate tissue samples in this study was approved by the Western Sydney Local Health District Human Research Ethics Committee [AU RED LNR/16/WMEAD/291 and AU RED HREC/17/WMEAD/63].

Prostate and ovarian cancer tissues were sampled using a 3 mm biopsy punch (Kai Industries, Seki City, Japan) and placed into barocycler microtubes (Pressure Biosciences Inc., South Easton, MA, USA). The wet weight of the prostate and ovarian cancer tissue samples ranged 0.7–14.4 mg and 0.3–6.9 mg, respectively. Prior to lysis, prostate cancer tissue punches were washed in barocycler microtubes to remove the OCT. In all, 150 μL of 70% (v/v) ethanol in Milli-Q water was added to the microtubes. The microtubes were then shaken at 2000 rpm for 1 min, followed by centrifugation for 2 min ($18000 \times g$, 4 °C). The supernatant was removed and 150 μL of Milli-Q water was then added, and the microtubes were shaken and centrifuged using the same conditions as above. The ethanol and water washing steps were repeated at least twice. Following lysis, digestion and solid-phase extraction (SPE) purification (described below), tissue samples were pooled to create separate ovarian cancer tissue and prostate cancer tissue samples.

**Preparation of yeast cultures**. Yeast cultures were sourced from a fully sequenced strain BY4741 (ATCC 4040002). The cultures were developed, selected and isolated using the streak plate method on an agar plate under aseptic conditions. The plate was incubated for 24 h after streaking, and then colonies were selected and isolated for culturing. Selected cultures were inoculated in Yeast Extract-Peptide-Dextrose (YPD) media (10 g peptone, 5 g yeast extract and 10 g glucose in 500 mL of water) and cultured using an orbital shaker incubator (33 °C, 200 rpm) for 72 hours. Yeast cells were harvested by centrifugation and the medium was removed. The yeast cell pellets were washed with phosphate buffered saline (PBS) and stored at −80 °C. A 3-mm biopsy punch was used to transfer yeast samples from the frozen cell pellets into barocycler microtubes. The wet weight ranged from 1.0 to 23.2 mg.

**Preparation of HEK293T control cell line samples**. HEK293T cell line was obtained from and authenticated by Cell Bank Australian using PCR (Promega kit PCR-PP16HS). Beginning with an authenticated sample, the cells were then expanded and then a bank of several frozen vials was generated for long-term storage. The cultures are generally terminated after ~30 passages. HEK293T cells were adapted to grow in Freestyle 293 Expression Medium (Life Technologies, Scorsby, VIC, Australia) supplemented with 200 mg/L G418 (Life Technologies) using a humidified shaker incubator (37 °C, 5% $CO_2$, 130 rpm). The adapted HEK293T culture was maintained in Erlenmeyer shaker flasks and scaled up in a 20 L WAVE bioreactor (GE Healthcare, Silverwater, NSW, Australia). The culture was seeded at an initial working volume of 5 L at a concentration of $0.8 \times 10^6$ viable cells/mL, with Pluronic F68 (Life Technologies) added at 0.2% (w/v) final concentration to reduce shear stress. As the concentration reached $3 \times 10^6$ viable cells/mL, the culture was scaled up to 20 L and maintained at 37 °C with a rocking speed of 25 rpm and rocking angle of 9°. The cells were harvested at $5 \times 10^6$ cells/mL by centrifugation ($1500 \times g$, 10 min, 4 °C), snap-frozen on liquid nitrogen and stored at −80 °C. Frozen cell pellets were scraped using a small spatula into barocycler microtubes. The wet weight ranged from 8.2 to 21 mg.

**Peptide extraction and preparation of sample mixtures**. The prostate cancer tissue, ovarian cancer tissue, yeast cells and HEK293T cells were each prepared as a single batch using Accelerated Barocycler Lysis and Extraction (ABLE) in a barocycler instrument, as described[38]. Specifically, tissue lysis, reduction and alkylation were carried out simultaneously using 31.5 μL of 1% (w/v) sodium deoxycholate (SDC), 5% (v/v) N-propanol, 100 mM triethylammonium bicarbonate (TEAB), 10 mM tris(2-carboxyethyl)phosphine (TCEP) and 40 mM iodoacetamide (IOA) in a Barocycler 2320 EXT (Pressure Biosciences Inc.) for 60 cycles

of 50 s at 45 kpsi and 10 s at atmospheric pressure, at 56 °C. Following this, 120 µL of rapid digestion buffer (Promega, Alexandria, NSW, Australia) and 1 µg of rapid trypsin/Lys-C (Promega) were added to each sample. Digestion was carried out in the barocycler using 30 cycles of 50 s at 45 kpsi and 10 s at atmospheric pressure, at 70 °C. Samples were acidified with formic acid (FA) to precipitate the SDC, then centrifuged for 15 min (18,000 × g, 4 °C). The supernatants for each sample type (ovarian cancer tissue, prostate cancer tissue, yeast cells and HEK293T cells) were pooled and cleaned up using Oasis HLB 3 cc (200 mg sorbent) SPE cartridges (Waters, Rydalmere, NSW, Australia). Each cartridge was activated with 500 µL of 90% (v/v) acetonitrile (ACN) in 0.1% (v/v) FA and then washed with 1 mL of 0.1% FA. Samples were diluted in 0.1% FA and loaded onto the cartridges. The cartridges were washed twice with 1 mL of 0.1% FA. Peptides were eluted with 300 µL of 70% ACN in 0.1% FA followed by 200 µL of 90% ACN in 0.1% FA. The eluates were pooled, dried, resuspended in 0.1% FA and then the concentration was determined using A280 nm with an Implen NanoPhotometer NP80 (Implen, München, Germany).

Using A280 nm quantitation, prostate cancer tissue, ovarian cancer tissue, yeast and HEK293T digests were diluted to 1 mg/mL using 0.1% FA. Different volumes of the diluted digests were then added to generate the desired sample compositions (Fig. 1a). The samples were also spiked with indexed retention time (iRT) calibration peptides (1 µL per injection) (Supplementary Table 1). All tubes were dried and stored at −80 °C. Prior to LC-MS/MS analyses, each tube was resuspended in 0.1% FA.

**Spectral reference library generation.** To decode sequential windowed acquisition of all theoretical fragment ion spectra (SWATH) spectra, an independently generated spectral reference library produced in data-dependent acquisition (DDA) mode is required. Individual spectral libraries were run on the mass spectrometer for prostate cancer tissue, ovarian cancer tissue and yeast. For each sample type, 100 µg of digest was dried and resuspended in 55 µL of 0.1% ammonium hydroxide for individual pre-fractionation. Pre-fractionation was carried out using a Shimadzu Nexera X2 LC-30AD HPLC equipped with inline DGU-20A 3 R and DGU-20A 5 R degassing units, SIL-30AC autosampler, CTO-29A column oven, SPD-M30A diode array detector and FRC-10A fraction collector. An XBridge C18 3.5 µm 2.1 × 150 mm column (Waters) was used for separation, with the column temperature maintained at 40 °C. The flow rate was 0.3 mL/min. The gradient started at 97% solvent A (10 mM ammonium hydroxide in Milli-Q water) and 3% solvent B (10 mM ammonium hydroxide in ACN) and was held constant for 3 min. The concentration of solvent B increased to 45% over 47 min and was held at 45% for 10 min. Peptides were eluted over the 60 min period and 60 separate 300 µL fractions were collected at 1 min intervals. Spectra were collected from wavelengths 190–600 nm with 214 nm used to determine peptide elution times. These fractions were then pooled into 15 fractions using a concatenation strategy, whereby every fifteenth fraction was pooled, as described[39]. The 15 fractions were dried and resuspended in 0.1% FA and 1 µL of iRT standards were added to each fraction.

The fractions were analysed by liquid chromatography-tandem mass spectrometry (LC-MS/MS) in DDA mode. The LC-MS/MS system setup was the same as for SWATH acquisition described below. The parameters were set as follows: ion source gas 1 (GS1) 20; ion source gas 2 (GS2) 20; curtain gas (CUR) 25; temperature (TEM) 100 °C; ion spray voltage floating (ISVF) 5500. The 90-minute DDA run consisted of a survey scan of 200 ms (TOF-MS) in the range 350-1250 m/z to collect MS1 spectra and the top 40 precursor ions with charge states from +2 to +5 were selected for subsequent fragmentation with an accumulation time of 50 ms per MS/MS experiment for a total cycle time of 2.3 s and MS/MS spectra were acquired in the range 100–2000 m/z.

The wiff format files generated in DDA mode were then converted to mgf format using the MSconvert tool from the ProteoWizard[40] package [version 3.0.18135 (79c747f66)]. The mgf files were searched against the UniProt[41] database (20,349 canonical sequences) appended with iRT peptides and decoy sequences (created by reversing the peptide sequences). Three search engines, Mascot[18] [version 2.6], X!Tandem[19] [version 2015.12.15.2] and MSGF+[20] [version 2018.09.12] were used for independent searches. The following search parameters were used for all three search engines: parent mass error was set to 50 ppm and MS/MS tolerance was set to 0.05 Da. Carbamidomethylation (C) was a fixed modification, while dynamic modifications were oxidation at methionine, deamidation at asparagine and glutamine, and acetylation at the N-terminal. Up to two missed cleavages were allowed for each fully tryptic peptide.

Results from Mascot[18] (in dat format), X!Tandem[19] (in xml format) and MSGF+[20] (in mzid format) were merged using PeptideShaker[21] [version 1.16.37] at a false discovery rate (FDR) of 1% at peptide-spectrum match (PSM), peptide and protein levels. The merged mzid file was then imported into Skyline[22] [version 4.2.0.18305] to create the spectral reference library. The library was cleaned for redundant spectra. Any modifications other than carbamidomethylation were excluded, and only peptides mapping uniquely to a protein were retained. iRT normalization was performed using endogenous iRT peptides.

The spectral library was further filtered to remove precursors less than 400 m/z, and product ions within 10 Da of the precursor. Minimum and maximum numbers of transitions per precursor were set to 3 and 6, respectively. The Skyline[22] spectral library was then exported to OpenSwath[23] format. Decoy transitions were added by reversing the sequences using OpenSwathDecoyGenerator from msproteomicstools

[version 2.4.0-HEAD-2018-12-09]. The library was converted to tsv and TraML formats using OpenSwathAssayGenerator from msproteomicstools to analyse with OpenSwath[23]. The resulting library contained 31,113 precursors from 6805 proteins, of which 4980 are human proteins. The unfiltered spectral library has been deposited to the ProteomeXchange Consortium via the PRIDE partner repository with the dataset identifier PXD015912.

**DIA-MS data acquisition.** All mass spectrometry (MS) runs were performed on an Eksigent nanoLC 415 high performance liquid chromatography (HPLC) operating in microflow mode, coupled to a TripleTOF® 6600 mass spectrometer from SCIEX™. Samples (2 µg in 5 µL) were injected onto a C18 trap column (SGE TRAPCOL C18 G203 300 µm × 100 mm) and desalted for 5 min at 10 µL/min with solvent A (0.1% FA). The trap column was switched in-line with a reversed-phase capillary column (SGE C18 G203 250 mm × 300 µm ID 3 µm 200 Å) that was maintained at 40 °C. The flow rate was 5 µL/min. The gradient of solvent B (99.9% ACN in 0.1% FA) started from 2% and increased to 10% over 5 min, 10 to 25% over 60 min, 25 to 40% over 5 min, 40–95% over 4 min, held at 95% for 5 min, followed by a 9 min column equilibration step with 98% solvent A. The liquid chromatography eluent was analysed using the TripleTOF® 6600 system equipped with a DuoSpray source and a 50 µm internal diameter electrode controlled by Analyst 1.7.1 software.

Peptide spectra were acquired in SWATH mode using 100 variable windows, as per SCIEX technical notes. The parameters were set as follows: lower m/z limit 350; upper m/z limit 1250; window overlap (Da) 1.0; CES was set at 5 for the smaller windows, 8 for the larger windows and 10 for the largest windows. MS2 spectra were collected in the range 100–2000 m/z for 30 ms in high resolution mode and the total cycle time was 3.2 s. The average peak width was 40.87 s, meaning that an average of 12.8 data points were acquired per peak.

On days when the experimental design did not require data collection, the six mass spectrometers generated data in DIA-MS mode for a diverse selection of samples from other studies. The instruments were not operating during cleaning times, but any other instrument downtime during the experimental period was negligible. A major instrument clean was defined as cleaning of the Q-jet, Q0, Q1 and TOF entrance lens. A minor clean was defined as a Q-jet clean, or Q-jet with Q0 clean only. Dates of instrument maintenance, including probe, analytical column and trap column changes are shown in Supplementary Fig. 1b.

**DIA-MS data processing pipelines.** The data processing pipeline used in this study is summarised in Supplementary Fig. 1c. Data in wiff file format were collected for 1560 MS runs. Two runs were excluded from downstream analyses due to error during acquisition by the mass spectrometer (Supplementary Data 1). The remaining 1558 data files were converted from wiff to mzML format with ProteoWizard[40] [version 3.0.18135 (79c747f66)]. These files were then analysed with OpenSWATH[23] [version 2.4.0, revision a7b4f64], implemented using the Docker container cmriprocan/openms:1.2.4. One run was removed after OpenSWATH processing due to inability to perform retention time alignment (Supplementary Data 1). PyProphet[24] [version 2.0.4] was used for FDR control, implemented using the Docker container cmriprocan/openms-toffee:0.14.2. We used a subsample ratio equal to 10/1557 and the software was run in non-parametric mode with scoring of both MS1- and MS2-level data. We applied a threshold of 5% run-specific peak group FDR cut-off, with a threshold of 1% for both global peptide and global protein FDR filtering. The code used to execute OpenSWATH and PyProphet is recorded in Supplementary Note 2. The data processing pipeline returned data from 1553 runs as no peptides were detectable after PyProphet filtering in four files (Supplementary Data 1).

For peptides with multiple charge states, MS2 intensity values of the corresponding precursors were summed to give peptide intensities. The spectral reference library was generated across two species and so, after processing the data independent acquisition (DIA) data files, any quantified peptides that were not uniquely mapping across the human and yeast proteomes combined were removed (n = 126 peptides). Samples were then excluded if they produced poor quality or inconsistent output. To do so, any samples with a mean Pearson correlation with replicates from the same sample of <0.9 were identified. Any samples in which peptides were observed from less than a threshold of 1200 proteins were also identified. In total, 26 poor-quality samples were excluded from the experimental dataset, leaving 1527 samples for downstream analysis (Supplementary Data 1). Peptides were next identified that were inconsistently observed across the study. To do so, peptides were retained if they were observed in more than 10% of replicate samples from at least one of Samples 1–7. In total, 934 peptides were excluded and 17,054 peptides were retained (including 29 iRT peptides used for retention time calibration; Supplementary Table 1) from 2796 proteins.

**Normalisation to remove unwanted variation.** We have developed a method for data normalisation based on RUV-III[28] (remove unwanted variation). The linear statistical model underlying RUV-III for data on n peptides from m DIA-MS acquisitions takes the form:

$$Y = MX\beta + W\alpha + \varepsilon \qquad (1)$$

where $Y$ is the $m \times n$ of matrix of observed peptide measurements on the log₂-intensity scale. On the right-hand side, $M$ would normally be the known $m \times m_1$

binary matrix with rows corresponding to acquisitions and columns corresponding to $m_1$ distinct biological samples, indicating which acquisitions are from which biological samples. X is the (in principle known) $m_1 \times p$ design matrix whose columns capture the biological variation of interest across samples. $\beta$ is the unknown $p \times n$ matrix of effects of the biological conditions on the peptide measurements. W is the unknown $m \times k$ matrix whose k columns capture the factors determining unwanted variation, $\alpha$ is the $k \times n$ matrix of effects of these unwanted variation factors on the peptide measurements, and $\varepsilon$ is the $m \times n$ matrix of error terms. The number k of columns of W is a key component of the model, and this too is unknown. For the goal of normalization, $X\beta$ plays no explicit role and can be considered to be biological signal in the data (in this case the human tissue dilution series), which is to be preserved. Our goal is to estimate the unwanted variation term $W\alpha$ and subtract it from Y without any impact on the term $X\beta$. This unwanted variation is sometimes called a batch effect, but in what follows it need not be restricted to the effects of known batches.

Key to success of RUV-III as a normalization strategy is the specification of technical replicates of each type of sample and also a suitably large number of negative controls (in this experiment, negative controls will be peptides). Negative controls are peptides whose measurements are known, or expected to be, unaffected by biological variation in the data (that is, they are not part of the biological signal) but will still be affected by unwanted variation. There are no pre-defined rules regarding the choice of negative controls, and domain-specific knowledge and human judgement are most important[28,42,43]. Endogenous genes will generally be a better choice of negative controls than will spike-ins, because the former reflect more unwanted variation in the experiment. For detailed discussion regarding how best to choose negative controls, and the optimal numbers of negative controls to use, readers should refer to references[28,42,43].

The premise of RUV-III is that differences of all measurements on technical replicates constitute unwanted variation, as does variation in the measurements on negative control peptides. RUV-III extracts information on unwanted variation from both of these sources and forms an estimate of $W\alpha$ which is then subtracted from Y. RUV-III requires the estimation of the dimension k of unwanted variation. In typical applications of RUV-III, k has been determined using information such as measurements on positive control peptides (when differential abundance is the goal), graphical displays or by exploiting some special context-dependent feature of the study. For this study we chose the value of k to be 20, as this gave a high median Pearson correlation between all human peptide intensities and ovarian cancer tissue proportion across the dilution series, as reported.

We have named the modification made to RUV-III as RUV-III-C (Remove Unwanted Variation III Complete) to denote the fact that we apply RUV-III only to subsets of complete data in a data matrix for a given experiment. RUV-III would normally begin by averaging technical replicates and passing to residuals (for details see Supplementary Methods in reference[28]). This is a suitable approach when all the measurements are $\log_2$-transformed non-zero intensities. However, this is not satisfactory when a dataset contains missing values that are a combination of true missing (that is, peptides that were in reality absent from a sample) and either missing not at random (MNAR) or missing completely at random (MCAR). It is clearly problematic to average together peptide intensities from a set of technical replicates that contain one or more large non-missing intensities along with one or more missing values. We could proceed by restricting ourselves to only those peptides present with non-missing intensities in all acquisitions, but this results in far too drastic a reduction in the number of peptides available for examination and this is not suitable for cohorts of heterogenous tissue types where true missing are expected. The use of familiar data imputation methods such as k-nearest neighbours could increase the numbers of peptides available for analysis, but this also brings additional risks and potential inaccuracies into the dataset. For this reason, we developed RUV-III-C, whereby we normalise the data on any given peptide p by applying RUV-III to only those acquisitions for which the measured intensities on p and all negative control peptides used are non-missing. After normalising the non-missing data in peptide p in this way, the missing values can be returned to where they were to complete the normalization. In this manner, we allow the missing values in peptide p to play little role in normalisation. We will execute as many runs of RUV-III on different subsets of acquisitions as there are peptides that we wish to normalise. Unless a dataset is extremely large, we do not expect computational resources to be a limiting factor.

RUV-III does not require that data from all samples be acquired with technical replicates. However, in the present study we need to be particularly careful with what we designate as technical replicates if we wish to obtain results that will apply to other DIA-MS studies, as these would typically have many more distinct samples and many fewer technical replicates. Hence, in order to define the matrix M in our present context for RUV-III-C, we modified the notion of 'distinct sample type'. To do so, we randomly partitioned the technical replicates of each of the eight sample types into groups of triplicates acquired on different instruments, and we then declared as technical replicates only those acquisitions corresponding to one of these groups. In effect, we have altered our definition of technical replicates in order to mimic a real-world scenario, whereby we normalise data with RUV-III-C using technical replicates acquired on different instruments at varying times from last clean.

This experimental design comprised eight sample types, with data for each obtained from many replicate MS runs. Seven of these sample types contained a variable yeast-prostate-ovary mixture. For this reason, there were no endogenous

peptides that could be regarded as negative controls in the usual sense. Instead, we obtained negative control peptides by scaling peptides from yeast proteins on the $\log_2$-scale within each sample type group, so that the mean intensity is identical between different sample types. Thus, we obtain a set of peptide measurements that satisfy the requirements of control variables for RUV-III. All we need to apply RUV-III-C satisfactorily for peptide p is that in the acquisitions that give a non-missing intensity for peptide p, the negative control yeast peptides also give non-missing intensities. For each peptide, we used variable numbers of negative controls up to a total of 1,622 peptides, which was the total number of peptides derived from yeast proteins that were identified in our study. In order for RUV-III with k factors to be successfully applied to a data matrix Y, the rows of M less columns of M must be $\geq k$. This condition must hold for each peptide, and so peptides were only retained where the number of runs recording non-missing intensities less the number of biological samples in which a non-missing intensity occurs was $\geq k$.

**Data analysis and interpretation.** Median normalisation was implemented using the medianNormalization function in the NormalyzerDE[44] package [version 1.0.0] in R[45] [version 3.5.2]. ComBat[29] was implemented using the sva[46] package [version 3.30.1] in R[45] [version 3.5.2], using median-normalised and natural scale data with default parameters. Batches were assigned according to which instrument was used for data acquisition. Missing values were set to zero for ComBat, and were returned as 'NA' values afterwards.

Principal component analysis (PCA)[25] was performed with the PCA function, using data that were standardised (mean equal to zero and standard deviation equal to one) by the StandardScaler function, both from the Scikit-learn[47] python package. The heatmap in Fig S2e was generated using the Clustermap function from the Seaborn[48] Python package. The heatmap colour range was computed with robust quantiles and the ward method, and missing values were filled with zeros. The predicted associations from linear regression in Fig. 3b, d were calculated using the OLS function from the statsmodels[49] Python package.

Frequently-observed peptides were defined as those peptides that were present in more than 80% of MS runs in each of Samples 1–7 ($n = 2950$ peptides). Ovarian cancer tissue-specific peptides ($n = 849$) were defined as peptides derived from human proteins that were present in more than 80% of runs containing ovarian cancer tissue but not prostate cancer tissue (Sample 7) and less than 5% of runs containing prostate cancer tissue but not ovarian cancer tissue (Sample 1).

For technical replacement, technical replicates were assigned as described above but each method used replicates spanning either two, three, four, five or six different instruments respectively. Missing values were replaced for a given peptide when a peptide had a non-missing intensity (a) at least once in a group of replicates of the same sample type, (b) at least twice in a group of replicates (for replicates spanning between three and six instruments), and (c) in more than half of a group of replicates (for replicates spanning between four and six instruments). Technical replacement was performed by replacing missing values with a value sampled from a normal distribution around the mean normalised non-missing intensity of the same peptide measured in technical replicates. For each technical replicate, a value was calculated for the standard deviation divided by mean (here termed log-CV) of each peptide within each set of technical replicates, then the mean of these log-CVs was taken. The standard deviation of the distribution used for technical replacement of a given peptide was set to 25% of the mean log-CV for that set of technical replicates, multiplied by the mean normalised peptide intensity. For these calculations, all values remained on the $\log_2$ intensity scale. The percentage of technical missing values replaced in the 3 MS ($\geq 2$) scenario (i.e., triplicates acquired on three mass spectrometers where replacement only occurred when a peptide was observed in two replicates) was reported as ~20%. This value was obtained by counting the number of missing values replaced in peptides that were non-missing in >10% of replicates, and the total number of missing values in these peptides before technical replacement. These values were calculated separately for each sample and were then added together, before dividing the totals to determine the overall percentage.

Likely true positive peptides were defined as those peptides observed in more than 90% of replicates of a given sample during the entire experiment period. Likely false positive peptides were defined as those peptides observed in less than 10% of replicates of a given sample during the entire experiment period. For each sample, likely true positive and likely false positive peptides were identified, and the number of values recorded (as a percentage of possible identification) was determined for each sample. The mean across all samples was reported in Fig. 4c.

For machine learning, we assumed that the underling relationship between ovarian cancer tissue proportion and the input peptides vector in the dataset was linear. For this reason, we used two supervised machine learning approaches. We first applied a simple regularised linear model, with LASSO (Least Absolute Shrinkage and Selection Operator) that can avoid overfitting in high dimensional data (results shown in Supplementary Fig. 8). We next applied a more complex multi-dimensional regression model known as a multilayer perceptron (MLP) regressor, which can better model any non-linear relationships (results shown in Fig. 6). MLP is a neural network based regression technique that is widely used for its flexibility, robustness and ability to model both linear and non-linear relationships between inputs and outputs[50,51]. The MLP regressor was used with five layers, namely an input layer (14,846 neurons), three hidden layers (10, 1000 and 10 neurons) and an output layer with a single neuron for producing real-numbered outputs. We used a

rectified linear unit activation function and a stochastic gradient descent optimiser for training, with L2 penalty parameter equal to 0.01 and a constant learning rate. Both the LASSO and MLP models were implemented with the Scikit-learn[47] Python package. In each case, the best alpha parameter (penalty term) was tuned for each training set separately, using grid search cross validation from Scikit-learn[47]. The grid search function used ten-fold cross validation for evaluation of the model during the tuning process. All remaining parameters were set as default.

Unless otherwise stated, data were analysed using custom Python scripts. In all cases, the threshold for significance was defined at $P < 0.05$.

**Multi-laboratory study data analysis**. The Collins et al.[17] multi-laboratory study analysed samples comprising a dilution series of 30 stable isotope-labelled standard (SIS) peptides in a background of HEK293 cells. Samples from five different dilution mixtures were distributed to 11 laboratories worldwide for analysis via DIA-MS. The CVs of these 30 SIS peptides were compared before normalisation, after median normalisation (the method originally used by Collins et al.), and after normalisation by both RUV-III and RUV-III-C.

To perform median normalisation, the median $\log_2$-transformed peptide intensity was computed for each run, and then subtracted from each peptide intensity for that run. Intensities for peptides not detected in a given run were marked as NA and did not contribute to the calculation of the median intensity. For normalisation by RUV-III and RUV-III-C, peptide intensity data were log-transformed and of the 8,555 human peptides that were detected in every sample, the first 500 peptides were held out to assess the quality of the normalisation and the remaining 8,055 were used as negative control peptides for normalisation. Technical replicates were formed by randomly grouping together pairs of MS results from the same mixture generated at different sites. Where such pairs could not be generated, some DIA-MS runs were treated as un-replicated. RUV-III and RUV-III-C were both applied with $k = 11$. For some peptides, RUV-III-C did not produce a normalised value as there were insufficient non-missing values to apply RUV-III-C with $k = 11$. The data were exponentiated before the CV was calculated.

**Reporting summary**. Further information on research design is available in the Nature Research Reporting Summary linked to this article.

## Data availability

The mass spectrometry proteomics data including the spectral library, have been deposited to the ProteomeXchange Consortium via the PRIDE[52] partner repository with the dataset identifier PXD015912. All other data are available from the corresponding author on reasonable request. Source data are provided with this paper.

## Code availability

The commands used to process DIA-MS acquisitions through OpenSWATH[23] and PyProphet[24] are recorded in Supplementary Note 2. The RUV-III-C implementation is available as an *R* package stored at the GitHub: https://github.com/CMRI-ProCan/RUV-III-C.

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

## Acknowledgements

Ovarian cancer tissue was accessed from the Gynaecological Oncology Biobank (GynBio-bank) Western Sydney Local Health District, Westmead. GynBiobank is a member of the Australasian Biospecimen Network-Oncology group funded by National Health and Medical Research Council (NHMRC) of Australia Enabling Grants 310670 and 628903 and Cancer Institute New South Wales (NSW) Grants 12/RIG/1-17 and 15/RIG/1-16. Assistance from Dr Maiken Marcker Espersen of the Westmead Institute for Medical Research and Catherine Kennedy of GynBiobank is gratefully acknowledged. Prostate cancer tissue was accessed from University Hospital Zurich, Switzerland. Assistance from Dr Christian D. Fankhauser (University Hospital Basel), Dr Thomas Hermanns, Dr Cédric Poyet and Dr Niels J. Rupp (University Hospital Zurich) is gratefully acknowledged. The provision of yeast by Prof Marc Wilkins of the University of New South Wales is gratefully acknowl-edged. ProCan® is supported by the Australian Cancer Research Foundation, Cancer Institute NSW (2017/TPG001, REG171150), NSW Ministry of Health (CMP-01), University of Sydney, Cancer Council NSW (IG 18-01), Ian Potter Foundation, the Medical Research Futures Fund (MRFF-PD), NHMRC of Australia (GNT1170739), and National Breast Cancer Foundation (IIRS-18-164). R.C.P. and P.J.R. are supported by NHMRC Fellowships (GNT1138536 and GNT1137064, respectively).

## Author contributions

R.C.P., P.G.H., R.S., Y.L., R.R.R., P.J.R. and Q.Z. were responsible for study design and/or conceptualisation. P.G.H., N.L., D.X., J.M.S.K., S.M., S.G.W., E.K.S., K.A., P.W., A.d.F. and R.B. were responsible for sample acquisition, sample preparation and/or mass spectrometry. R.C.P., R.S., M.W., M.H., M.D. and B.T. were responsible for Open-SWATH and PyProphet pipeline implementation and data processing. R.C.P., R.S., S.S.M., A.A., M.A.W., J.Y., R.A., T.S., Y.L. and Q.Z. were responsible for data analysis, interpretation, and/or statistical approaches. R.C.P., P.G.H., R.S. and Q.Z. wrote the paper with input from other authors. All authors read and approved the paper.

## Competing interests

K.A. is an employee of SCIEX, which operates in the field covered by the article. R.A. holds shares of Biognosys AG which operates in the field covered by the article. The remaining authors declare no competing interests.
