## [Peer Review File · Nature Communications]

Reviewers' comments:

Reviewer #1 (Remarks to the Author):

The manuscript by Poulos et al. titled "Enabling large-scale proteomics for reproducible research" describes a rather large-scale assessment of reproducibility across over 1500 data-independent acquisitions of pooled human cancer sample specimen. The study assesses technical replicate acquisitions of these pooled samples also across 6 different instruments and over several months. The study represents an interesting contribution that is relevant to future high-throughput clinical MS-based work. Additionally, some new software algorithms are presented addressing some issues identified during the study to improve overall reproducibility and data quality. This is a valuable contribution.

Comments:

- It is unclear how many proteins are identified and quantified in the DIA acquisitions in this study – this should be stated. In the methods sections the authors mention that > 6000 proteins are identified in DDA acquisitions (using the fractionation). But how many proteins are identified and quantified from the DIA acquisitions.
- In the DDA acquisitions the >6000 proteins – I assume that is the combination of how many proteins are identified from yeast and human (how many human proteins overall are identified?) – of those how many human proteins were identified and quantified from the DIA acquisitions?
- I assume there is a 2 peptide minimum applied per protein, that should be specifically stated.
- The way how the library is built should be explained in the main text also, not just in the methods.
- On page 6 the authors state that they measure an average of 6 peptides per protein. However, it would be more interesting to see for example a histogram indicating the distribution – i.e., how many proteins show 2 peptides, 3, 4, 5 etc.
- The authors state that some acquisitions were excluded that did not fulfill criteria of 'baseline quality control'. What are those criteria (mention in the main text).
- As different amounts of cancerous tissues as well as yeast are mixed in the various different samples – the authors should state if they observed any 'matrix effects'. Of different complexity within the samples. Did that influence the quality of the human identifications and quantification (better in the samples that have less yeast? or better in the samples that have more human material). Or is the matrix overall no factor here?
- In Figure 3a the authors describe some reproducibility behavior of a single human peptide, with and without normalization – what is that peptide? What protein does it originate from? PSA ? It

would be informative to have not just one single human peptide analyzed in this way (from higher/lower abundant proteins ..).

- There should be a supplemental table submitted as part of the supplemental material of this manuscript that has all the MS identification details from the search engine that was used to generate the spectral library.

- There should be a specific mention in the methods that the spectral library is provided as part of the data upload to Pride, including the Skyline file that the authors mention. That is an important resource as part of this manuscript and potentially of great interest for other scientists.

- The statistical processing and the algorithm RUV III c are very interesting – I understand it relies heavily on having peptides that do not change (negative controls). The authors mention in their case they choose yeast peptides as negative controls that do not change. Could the authors please elaborate or recommend how this should best be done for studies that are 100% human. The problem with using house keeping proteins is also to know for sure what is house keeping, what really does not change is sometimes difficult to assess. How was this done for the Bruderer et al study and for the Collins et al study – this would be important to detail more as this provides insights into how to apply this strategy in a broader way ...

- How many negative control peptides do you need to get good statistics for this approach – was that assessed. As the authors are using likely many yeast peptides this seems to be working fine – how many negative control peptides do the authors use and what is the minimal that does still work ?

- It would be important to know how this method RUV III c works overall for low abundant peptides – is that more challenging ?

- The technical replacement strategy is that dependent on the retention time stability. I know the authors use iRT standards – it would be worth to discuss this a bit more.

- Overall the authors focus mainly on the MS metrics and performance – was there an overall effect of chromatography performance ?

- Acquiring 3 technical replicates per sample is often rather challenging for laboratories and many laboratories do not have as many instruments available. Typically, as there is so much more variability in biological samples it has been considered having more biological replicates is beneficial. I do think this is a challenge of this particular design for many labs to implement. If one does have enough biological replicates would one not rather acquire data for the biological replicates instead of acquiring 3 technical replicates for each biological sample? I can rather see the point with small clinical cohorts as the authors point out.

- just reporting CV's as measure for reproducibility should be extended to also do some measurements of true quantification - forming ratios and reporting those. In this study design this is possible as different amounts of human material is in each individual sample. Figure 5 mentions some of that but rather than just reporting incidence of significance. Could the authors form some ratios at different abundance levels and see how those change or maintain significance. What is the robustness of quantification which is often much more important than a CV.

Reviewer #2 (Remarks to the Author):

In the current manuscript, Poulos et al report on the reproducibility of DIA-MS measurements on six Sciex QTOF instruments within the ProCan Proteomics facility. On the basis of 1,560 DIA runs using samples of controlled composition and acquired over a period of 4 months, the authors first determined instrument performance over time and used the data to develop novel methods for data normalization and missing value imputation that, according to the authors, substantially improved predictive accuracy of sample contents.

As proteomics is moving towards clinical application, questions regarding robustness and reproducibility have to be answered in order to be able to use such data appropriately. In that sense, the current study is timely and it likely is the largest such studies performed to date. The authors provide an honest account of their data which is very laudable, but this reviewer has a lot of questions regarding how the data is presented and does not always agree with the conclusions drawn. Overall, this study is useful but needs a lot of work on the text in order to make it clearer / more accessible for most readers. In addition, the overall results are quite expected, particularly in light of the fact that conceptually similar work has already been published (Collins et al, Mol Cell Proteomics 2017) which asks the question why it should be published in Nature Communications rather than in a more specialized journal.

Specific points:

Study design and data acquisition: The study design and Figure 1 is somewhat confusing. The sample numbering in panel 1a does not match the one in panel 1b. That aside, it is unclear why samples 1, 2, 7 and 8 were run in duplicate while samples 3, 4, 5 and 6 were run in triplicate (here panel 1a and 1b contradict each other regarding sample 2). Is there any particular reason for this other than this creates 20 samples? The use of triplicates throughout would have allowed for a better intra-batch assessment of reproducibility. In addition, it is not obvious why the scheduling of the QC was chosen as it was. Four replicates were performed at the beginning and the end of the study but the times in between were different. Did the authors chose this schedule on purpose? The main text states that "Mass spectrometer maintenance schedules varied according to each individual instrument's performance". Does that mean that Fig 1c is just a schematic representation and the actual QC runs happened at different schedules for different instruments? What do the authors mean by 'base line controls'? Please clarify. According to the main text, 80% of proteins were supported by evidence from at least 2 peptides. This seems unusual for a DIA measurement (nowadays, this figure is even reached by DDA) unless proteome coverage was very deep. But this does not appear to be the case. Any idea why that is? It would be useful to state in the main text roughly how many proteins/peptides were identified in the QC runs (say on average in each and

collectively in all and how many were shared between all QC runs of the same type) so that readers get a better idea about how many data points went into the analysis that follows.

Baseline data reproducibility during a single experimental week: For the reproducibility analysis of the short term QC samples, why was only the second set of samples used? Does the first set of samples show the same results? Fig. 2a: can you comment on what the 'streaks' in the PCA plots are? Do you already see that the performance within this short period of time shows a trend? The statement that "We observed an approximately linear relationship between the intensities of ovarian cancer tissue-specific peptides" is clearly not warranted. Fig 2c has no y-axis annotation but it is clear that the data follows a (shallow) exponential loss of intensity over the 'dilution series'. This is very surprising as the 'dilution' only spans a factor of 16. DIA is supposed to be very quantitative but this does not appear to be the case for this particular data. How do the authors explain the detection of PSA in sample 7 (no prostate) and the very large variability of the intensity of the four different PSA peptides? Is this carry-over? In addition, even though one can see an association of the intensity of TAR DNA-binding protein 43 and the proportion of the amount of ovarian tissue in the sample, the relationship is not linear. Does this perhaps imply that the QTOF detector already saturates at log₂ intensities of 13 or higher? This effect should be analysed more systematically. There are thousands of peptides for which this could be done. At this stage, it is preliminary to conclude that DIA-MS is generally reproducible.

Decrease in instrument sensitivity over time since cleaning: Fig. 2a and 2b show a ~4x variation of peptide intensities over just 4 weeks and the authors do point this out. Although this reviewer does not question the general validity of the statements made around Fig. 2b, I would be very careful when interpreting results of Pearson correlations that only have 4 data points (why was the 50ug sample not included?).

Development of a method for data normalization: It is not clear to this reviewer why the 'novel' normalization procedure was necessary or if it really improved the results. Looking at Fig 3c and the supplementary figures, it at least visually looked as if median centering the data produced better intensity normalized results than RUV-III-C. At least when using all peptides for normalization. Why do the authors show the normalization based on the iRT peptides only in the main figure? Again, one should be very careful not to over-interpret Pearson correlations on four data points as shown in Fig. 3d. What is quite puzzling is that the new normalization procedure did not really improve results for an individual instrument (Fig. 3f). Hence, the improvements across the entire data set is surprising in its extent.

Technical replacement of missing values: Missing value imputation is a tricky business and the authors may want to point this out. The Figure 4a is a good way of showing this but I would suggest to add a line that shows the intensity distribution of the high-confidence peptides to be able to see if MCAR peptides are similar or dissimilar to the frequently detected peptides. However, I don't think it is necessary to show six almost identical plots to make this point. One could show one or two and move the rest to the supplement. Similarly, it would suffice to show one or two instrument panels in Fig 4b. I suggest that the extra space generated in this way could be used to illustrate that/how imputing missing values improves reproducibility. This is not really captured in Fig.4c but would be important to show. The current Fig 4c raises the question as to how important the missing value imputation actually is. Even without any 'technical replacement', the true positive ISs are already at

>97% (at 5% FP). Increasing the true positives by just 2% comes at the cost of increasing false positives by almost a factor of two. The authors should therefore tone down the importance of this imputation (albeit acknowledging that this does not seem to hurt much).

Simulation of cohort analyses in proteomics: From reading the main text, it is not very clear what Fig. 5a shows. The same applies to Fig. 5b. While the reviewer can guess what the solid blue lines show, the grey areas and the lines in them are not clear. Perhaps I do not get the figure at all but the data in 25% ovarian cancer tissue proportion panel in Fig 5b implies that it was impossible to distinguish samples with 25% or 50% ovarian cancer content. This would be highly unexpected as this should be the easiest of all comparisons. Please clarify.

Prediction of tissue proportion in a complex mixture: It would indeed be exciting if this approach could work. However, the machine learning part is incomprehensible from reading the main text and, more importantly, how relevant is it to train such a classifier based on a mixture of two cancer types and yeast that are mixed into one sample. Such a situation would never occur when analyzing clinical specimen. Furthermore, in light of the data from the previous section, this reviewer seriously doubts that any such predictions can be meaningfully performed on the very limited amount of available data used for training. How do the authors make sure that the model is not entirely overtrained? In my opinion, a 10-fold cross validation does little/anything to validate the predictions. Instead, another data set should have been generated to test the performance of the trained classifier.

Other points:

This reviewer suggests that the authors came up with an alternative title that better reflects the content of the paper. The current one is too general and likely turns away a lot of potential readers.

There are a few minor issues with language and grammar that should be fixed.

The text could be clearer in places. E.g. in the abstract, it is not clear what 'control' is. This becomes clear in the main text (HEK293T cells). Also, the term 'dilution series' is not accurate for the description of samples 1-6. The total peptide content in these samples is the same but the proportions of the ovarian cancer and yeast samples are different.

Further comments by members of the reviewer's laboratory: these are provided for information only.

Co-worker I:

Major comments:

- In particular in the first paragraphs there is a lot of repetition of what is already known.

o For example, the reproducibility chapter was already covered by Collins, et al 2017 in a more comprehensive way (intra-day variation, inter-day variation, even inter-lab variation). Other studies focus on the reproducibility and repeatability of DDA data (Tabb et al, 2010).

o Another example is the sensitivity decrease post cleaning chapter. The results are common knowledge

o It would be more desirable if the entire data set is analyzed in terms of long-term reproducibility which is novel and interesting! In this case it would be helpful to name the factors that mostly influence reproducibility.

o The RUV-III-C algorithm is a variant of RUV-III. As cited in the manuscript, another group already implemented this algorithm for gene expression data. In the present study, the authors transferred this knowledge to proteomics data and extended the algorithm to data sets with missing values.

- One selling point of DIA is that it has in general fewer missing values than DDA. Why do the authors apply their method not to DDA data or show the difference between DDA and DIA. It would be great if the normalization procedures could be applied to all kind of proteomic data sets. This makes an even stronger point than comparing the results of two mass spectrometric platforms (Sciex vs Orbitrap), because there the output data is very similar.

- How long does it take to normalize data and replace missing values with ProNorM? Which computational resources are required? Do the authors attempt to include their procedure as an option in standard DIA analysis software tools? Is the data first normalized and then imputed or the other way around? Why was this order chosen (Karpievitch, 2012 show that the order matters)?

- It would be great if authors could show a biological example where this normalization and imputation strategy outperforms standard normalization and imputation strategies. Maybe there are examples where without these novel algorithms, no biological conclusion can be drawn.

Minor comments:

- Introduction and study goal: The study goal is much better outlined in the abstract than in the introduction. Major aim was not the evaluation of reproducibility, but the development of a new algorithm that improves reproducibility.

- page 6, paragraph 2 and methods section: `Many peptides were consistently identified...`

Filtering happened before. It would be also great to show how consistent the data is without filtering

- page 6, paragraph 3: Only quantitative evaluation, but what about the quality of the data, e.g. repeatability (similar to Tabb et al). What about intra-day, inter-day, inter-instrument analysis? Additionally, it would be very helpful to get the acquired data points per peak so that the reader can get a feeling of the quality of quantification.

- Page 6/7: What about chromatographic stability? easy check with iRTs, does this influence the identification and quantification of peptides?

- Page 7, paragraph 1 and 2: What about the abundance of the chosen peptides?

- Page 10, paragraph 1: 'We then replace missing values ... observed for that peptide in a technical replicate'. What happens if the replicates do not contain this peptide? This might be a peptide that is regulated and might be of interest or is a false identified one.

- Page 11, paragraph 2: what's the quality of the 25 replicate samples? From which time points were they taken (after cleaning, before and after cleaning, etc)? A comparison to standard normalization techniques like grand mean normalization would strengthen the conclusion. Comparison to non-normalized data and a conclusion that the algorithm improves statistical significance is obvious.

- Page 20, 515, 516: Precipitation of SDC, page 20: no washing

- Page 20, 530; Page 21, 548: Library was generated with 0.5 % FA, DIA measurements with 0.1% FA. Any reason for that?

- Page 21, 561: Why using three different search engines? What's the overlap between them?

- Page 22, 574: 'peptides (including modified peptides): Use precursors information or use the term modified peptides instead

- Page 22: 7.000 proteins from 39.033 peptides. What is total protein/peptide number without all the filtering?

- Page 23: 100 variable windows with 3.2 sec cycle time \approx dp/peak?

- page 24, 620: Why excluding samples with $R^2 < 0.9$ in replicates? This is a study about reproducibility. Why excluding files with less than 1.200 proteins, what's the explanation?

- Page 27, 695: What about propagation of variation when scaling? \approx Normalization is applied already to this set,

Coworker II

Key points in this manuscript

1. We demonstrate for the first time the reproducibility of large-scale DIA-MS measurements over extended periods and across multiple mass spectrometers.

This is true, but I personally doubt the value of this experimental design.

2. To reflect real-world operating conditions, approximately 6,500 MS runs were carried out over a four-month period, during which the mass spectrometers received minor and major maintenance on an as-needed basis.

I think this is not very meaningful, unless they aim to generate dataset under different MS sensitivity at the beginning. The sensitivity loss is well known in proteomics labs. In the real-world operating conditions, researchers can easily monitor the machine performance by QC sample, the maintenance can be very flexible based on the QC data instead of an as-needed basis. In such big project, QC samples must be run to monitor the real time system performance.

3. We developed a novel strategy for MS data processing called ProNorM, which uses negative controls and technical replicates to normalise data from multiple instruments, reducing technical variation while strengthening biological signal.

4. ProNorM rescues approximately 20% of values likely missing for technical reasons, by replacing these with plausible intensities from a distribution derived from replicates run on other instruments.

The ProNorM method was developed to distinguish the real missing values due to low intensity (there is a concentration series in sample 1-8, there will be a trend. If the peptides have intensity

values in high concentration, and NA in the low concentration, they call it complicatedly random NA) and missing values due to random reasons.

ProNorM method seems to rely on the technical replicates. However, in the real DIA/SWATH analysis, 1) it is time consuming to run lots of duplicates for each sample, 2) the experimental intensities of peptide are unknown within different individual samples. Therefore, the ProNorM method seems not to work for real samples.

5. DIA-MS data acquired over extended time periods from multiple instruments with differing maintenance schedules were reproducible, and changes in low-intensity peptide signals were detected in a dilution series comprising known ratios of human cancer tissues.

I do not know which data they used to conclude this. If it is Figure 2c, I will argue the “ovarian cancer-specific peptides” are mostly high abundant peptides in the ovarian tissue sample.

Real and relative proportion of different samples in each sample ID, generated to understand what they are doing.

Sample ID 1 2 3 4 5 6 7

Yeast (%) 50 46.875 43.75 37.5 25 0 50

Ovarian (%) 0 3.125 6.25 12.5 25 50 50

Prostate (%) 50 50 50 50 50 50 0

Yeast (%) 16 15 14 12 8 0 16

Ovarian (%) 0 1 2 4 8 16 16

Prostate (%) 16 16 16 16 16 16 0

Human

peptides amount 16P 16P + 1O 16P + 2O 16P + 4O 16P + 8O 16P + 16O 16O

Comments

1. Figure 1a/b and Figure 2. The appearance order of yeast/ovarian/prostate in the names of each sample is different, making it a little hard to understand the results.

2. Figure 1b. It is not very easy to understand the experiment design from Figure 1b, the numbers in 1a and 1b represent different meaning. Considering one 48h period serves as one independent cycle, a ring cycle may be a better way to show the workflow.

3. Figure 2a. It is interesting to see there are some data points of sample 1-7 spreading out the grouping area in Figure 2a. How about the PCA analysis results after data normalization?

4. Page 6, line 154: Figure 2c.

It is very important to show the intensity distribution of “frequently-observed human peptides” across sample 1-7, Figure 2e only show one example.

5. Page 6, line 158:

The definition of “Ovarian cancer tissue-specific peptide” is unclear.

According to my understanding, the definition of “Ovarian cancer tissue-specific peptide” might be:

1) the “Ovarian cancer tissue-specific peptide” were filtered out based on the human peptides identified in both sample 1 and sample 7, all the other datasets from sample 2/3/4/5/6 were not used.

2) In summary, there were 156 runs of sample 1 and 7, respectively. Therefore, the appearance of these “Ovarian cancer tissue-specific peptide” should be < 8 runs in sample 1 (not containing ovarian tissue), and > 125 runs in sample 7 (not containing prostate tissue).

It will be much easier to understand the results if the authors describe the definition in details.

6. Page 7, line 163: Figure 2d

The numbers of samples containing quantification information of PSA peptides in sample 1 and 6 are much lower than sample 2-5 in Figure 2d. This is due to the different DIA-MS runs of sample 1/6 (n=156), and sample 2-5 (n=234). In order to make the data much clearer, it is better to show the relative proportion in each sample instead of absolute numbers in Figure 2d.

7. Page 7, line 166: Figure 2e

As the house-keeping protein TARDBP is constitutively expressed, the theoretical mass ratio of TARDBP in sample 1/2/3/4/5/6/7 should be 16:17:18:20:24:32:16. Based on the sample design, the TARDBP intensity in sample 1 and 7 should be the same.

However, 1) the intensity of TARDBP in sample 7 is much higher than that of sample 1 in Figure 2e; 2) the number of samples containing TARDBP quantification information in sample 7 is also much higher than sample 1 in Figure 2e.

How to explain these differences?

8. Page 7, line 172: Instrument sensitivity

1) As shown in Figure 3a-b, the instrument sensitivity variation seems to be highly instrument specific. For example, “minor instrument clean” did not bring back the sensitivity of M02 and M06. However, the sensitivity gain of M05 after the “major instrument clean” is almost neglectful, the “minor instrument clean” already kept its sensitivity.

Data from Figure 3a demonstrates that QC run is very useful to monitor the instrument sensitivity. As the authors run lot of other samples alongside this project, it is curious for them not to monitor the instrument sensitivity by QC in such big project. It is more meaningful to schedule the instrument maintenances based on QC results.

2) The authors showed the MS2 level sensitivity data in Figure 3, it is very interesting to see the MS1 level sensitivity data. The reviewers sometimes found the sensitivity loss at MS2 level rather than MS1 level from the orbitrap type machine.

3) How about the quantification dynamic range at both MS1 and MS2 level in this study?

4) Many researches claim that the sensitivity of DIA/SWATH is higher than DDA. In the original MS1 spectra, there will be both identified and non-identified peaks. It is interesting to show the precursor ions intensity proportion of each identified peptide in the total intensity of peaks in the corresponding MS1 spectra. This analysis may give a clear view of the dynamic range of DIA/SWATH.

9. Page 174 and Figure 3b: The authors should clearly claim that the example human peptide was specifically selected from the “ovarian cancer tissue-specific peptides”.

10. Page 8, line 206: Figure 3f: The theoretically mass of “frequently-observed human peptides” in sample 1-7 is different. Although the authors described that the intensity CV of each peptide was calculated for each sample in the figure legend, it is still not very clear to understand the data. It is better to show some example plots illustrating the intensity CV of each peptide from different samples.

11. Page 11, line 291: Tissue proportion predication is pretty hard, as the real tissue-specific proteins and peptides should be used. In this study, the tissue proportion works well when the proportion difference of ovarian and prostate is very high, indicating the “Ovarian cancer tissue-specific peptide” dataset might be not very “tissue-specific”.

As the number of ovarian tissue specific proteins may be very small, it is interesting for the authors to reduce the numbers of “Ovarian cancer tissue-specific peptide”, for example, only keep the peptides show very good linear intensity relationship to ovarian tissue proportion. What about the predication performance of this predication tool then?

By the way, considering the yeast proteins will not be shared by ovarian and prostate tissue. How about the “tissue proportion” of yeast sample in sample 2-5?

12. Page 13, line 328: It is no doubt that increasing the number of technical replicates may be a useful strategy for analysis of small clinical cohorts. However, lots of replicates in DIA analysis lose its throughput advantage.

Methods:

13. The DDA data were searched independently with three different search engines to generate the spectral library. Are there some MS/MS spectra that are matched to different peptides by the three

search engines? If one MS/MS spectra was matched to different peptides, how to deal with this issue when combining the three results into one dataset?

Point-by-point Response to Reviewers' Comments

Strategies to enable large-scale proteomics for reproducible research

Authors:

Rebecca C. Poulos^{1#}, Peter G. Hains^{1#}, Rohan Shah^{1#}, Natasha Lucas¹, Dylan Xavier¹, Srikanth S. Manda¹, Asim Anees¹, Jennifer M. S. Koh¹, Sadia Mahboob¹, Max Wittman¹, Steven G. Williams¹, Erin K. Sykes¹, Michael Hecker¹, Michael Dausmann¹, Merridee A. Wouters¹, Keith Ashman², Jean Yang³, Peter Wild⁴, Anna deFazio⁵, Rosemary L. Balleine¹, Brett Tully¹, Ruedi Aebersold⁶, Terence P. Speed⁷, Yansheng Liu⁸, Roger R. Reddel¹, Phillip J. Robinson¹, Qing Zhong^{1*}

Affiliations:

¹ ProCan[®], Children's Medical Research Institute, Faculty of Medicine and Health, The University of Sydney, Westmead, NSW, Australia.

² Sciex, 2 Gilda Court, Mulgrave, VIC, Australia.

³ School of Mathematics and Statistics, The University of Sydney, Sydney, Australia.

⁴ Dr. Senckenberg Institute of Pathology, University Hospital Frankfurt, Frankfurt am Main, Germany; and Department of Pathology and Molecular Pathology, University Hospital Zurich, Zurich, Switzerland.

⁵ Centre for Cancer Research, Westmead Institute for Medical Research, Westmead, NSW, Australia; Faculty of Medicine and Health, The University of Sydney, Westmead, NSW, Australia; and Department of Gynaecological Oncology, Westmead Hospital, Westmead, NSW, Australia.

⁶ Department of Biology, Institute of Molecular Systems Biology, ETH Zürich, Zürich, Switzerland; and Faculty of Science, University of Zürich, Zürich, Switzerland.

⁷ Bioinformatics Division, Walter and Eliza Hall Institute of Medical Research, Parkville, VIC, Australia; and Department of Mathematics and Statistics, University of Melbourne, Melbourne, VIC, Australia.

⁸ Department of Pharmacology, Yale University School of Medicine, New Haven, CT, USA; and Yale Cancer Biology Institute, Yale University, West Haven, CT, USA.

Notes:

These authors contributed equally; * Correspondence.

Reviewer #1

Comments:

1.1 It is unclear how many proteins are identified and quantified in the DIA acquisitions in this study – this should be stated. In the methods sections the authors mention that > 6000 proteins are identified in DDA acquisitions (using the fractionation). But how many proteins are identified and quantified from the DIA acquisitions.

This detail was provided in the *Methods* section of the original manuscript and has now been added to the *Main Text* of the revised manuscript (underlined):

*“... 1,527 runs were retained (see **Methods**; **Supplementary Table 1, 2**). These files comprised data from a total of 17,054 peptides derived from 2,796 proteins, of which 2,363 were human proteins.”*

-- Results, Study design and data acquisition, Para. 2

1.2 In the DDA acquisitions the >6000 proteins – I assume that is the combination of how many proteins are identified from yeast and human (how many human proteins overall are identified?) – of those how many human proteins were identified and quantified from the DIA acquisitions?

Of the 6,865 proteins that comprised the spectral library, 5,013 are human proteins. This information has been provided in the revised manuscript, as below (underlined):

“The resulting library contained 31,614 precursors from 6,865 proteins, of which 5,013 are human proteins.”

-- Methods, Spectral reference library generation, Para. 4

There were 2,363 human proteins identified and quantified and this information is now provided in the revised manuscript – see Response 1.1.

1.3 I assume there is a 2-peptide minimum applied per protein, that should be specifically stated.

When quoting the overall numbers of proteins, there was no 2-peptide minimum applied. Instead, the proportion of proteins quantified by at least two peptides is specified in the text. Proteins were only identified by unique peptides, and we now note this in the revised manuscript, as below (underlined):

“All proteins were identified by unique peptides (an average of 6.1 unique peptides were identified per protein), and 80% of proteins were supported by evidence from at least two peptides (Supplementary Fig. 2a).”

-- Results, Study design and data acquisition, Para. 2

Further information regarding the number of peptides per protein is now included in a revised *Supplementary Figure 2a*, shown in Response 1.5.

1.4 The way how the library is built should be explained in the main text also, not just in the methods.

We now provide additional details (underlined) regarding spectral library generation in the *Main Text* of the revised manuscript:

“A spectral library was generated by merging search results from three search engines (Mascot³⁷, X!Tandem³⁸ and MSGF+³⁹) using PeptideShaker⁴⁰ and Skyline⁴¹ (see Methods). The MS raw data files were then processed with reference to this spectral library using OpenSWATH¹⁸ and PyProphet¹⁹ with 1% FDR at both global peptide and protein levels for identification and 5% run-specific peak group FDR for quantification (Supplementary Fig. 1c; see Methods).”

-- Results, Study design and data acquisition, Para. 2

Further information is provided in the *Methods*.

1.5 On page 6 the authors state that they measure an average of 6 peptides per protein. However, it would be more interesting to see for example a histogram indicating the distribution – i.e., how many proteins show 2 peptides, 3, 4, 5 etc.

This information was provided as cumulative data in *Supplementary Figure 2a* in the original manuscript. To improve clarity and provide more information as requested, we now also include a histogram in the revised figure that displays non-cumulative data, as shown:

Supplementary Figure 2. Distribution of non-normalised SWATH-MS data. (a) Cumulative (blue; left axis) and actual (red; right axis) numbers of peptides supporting each protein identification. Blue dotted lines indicate the point at which the cumulative number of proteins have support from at least two peptides.

1.6 The authors state that some acquisitions were excluded that did not fulfil criteria of ‘baseline quality control’. What are those criteria (mention in the main text).

In addition to detail about quality control in the *Methods*, some basic descriptive information (underlined) has been added to the *Main Text* in the revised manuscript:

*“After removing files (n = 26) that did not pass baseline quality controls such as total numbers of proteins identified and correlation with replicate samples (see **Methods**), and discarding a small number of peptides that were inconsistently observed across the cohort (n = 934; see **Methods**), 1,527 runs were retained (**Supplementary Table 1, 2**). These runs comprised data from a total of 17,054 peptides derived from 2,796 proteins, of which 2,363 were human proteins.”*

-- Results, Study design and data acquisition, Para. 2

1.7 As different amounts of cancerous tissues as well as yeast are mixed in the various different samples – the authors should state if they observed any ‘matrix effects’. Of different complexity within the samples. Did that influence the quality of the human identifications and quantification (better in the samples that have less yeast? Or better in the samples that have more human material). Or is the matrix overall no factor here?

We did not observe any significant matrix effects in the samples analysed in our experiment.

The potential for sample matrix complexity to impact protein identification was investigated by examining the mean proportion of samples across the dilution series in which n = 1,000 of the highest and lowest intensity human peptides from a 100% human tissues sample (Sample 6) were detected. Consistent with an expected dosage effect, we found that some of the lowest intensity peptides were inconsistently observed as less human tissue was present in a sample (see *Revision Figure 1, upper panel*). However, if a matrix effect was present, we would expect to find that even some of the highest intensity peptides would also be inconsistently observed across the dilution series due to ion suppression effects. This did not occur (see *Revision Figure 1, lower panel*), suggesting that matrix effects did not impact our ability to identify peptides that should otherwise be easy to observe.

Moreover, we did not observe any matrix effects relating to our ability to quantify peptides across samples of different complexity (see *Supplementary Figure 2g*).

Revision Figure 1 – Mean percentage of samples in which peptides were identified for $n = 1000$ of the highest (upper) and lowest (lower) intensity peptides observed in Sample 6.

1.8 In Figure 3a the authors describe some reproducibility behaviour of a single human peptide, with and without normalization – what is that peptide? What protein does it originate from? PSA? It would be informative to have not just one single human peptide analysed in this way (from higher/lower abundant proteins).

Figure 3a depicts $n = 29$ indexed retention time (iRT) peptides (see Supplementary Table 3). This information was provided in the figure title and legend in the original manuscript. Supplementary Figure 3a in the original manuscript depicts all peptides.

We observed the same trend relating to peptide intensity and instrument performance degradation in both the highest intensity ($n = 10$) and lowest intensity ($n = 10$) iRT peptides (see Revision Figure 2). We conclude that regardless of peptide intensity, the longitudinal trends that we describe in our study, and that are associated with instrument performance degradation, are the same.

Highest intensity:

Lowest intensity:

Revision Figure 2 – Intensities of $n = 10$ of the highest (upper) and lowest (lower) intensity indexed retention time calibration peptides before normalisation. Boxplots are coloured by instrument, within which data are ordered from earliest experimental day (left) to latest experimental day (right). Maintenance schedules of major (red) and minor (blue) instrument cleaning are indicated. Data are shown for replicates of Sample 5 and only every sixth experimental day is labelled on the horizontal axis.

1.9 There should be a supplemental table submitted as part of the supplemental material of this manuscript that has all the MS identification details from the search engine that was used to generate the spectral library.

This information is available via the PRIDE repository in the *Result* file entitled *Poulos_et_al_Peptide_intensity_matrix*. We have also now uploaded to the PRIDE database the Skyline and identification files requested, which relate to the spectral library searches. Details for accessing our PRIDE data deposition (PXD015912) are provided in the manuscript. Please note that we have chosen not to include these data as *Supplementary Information* because the files are large, and we feel that PRIDE is the most appropriate repository for this information.

1.10 There should be a specific mention in the methods that the spectral library is provided as part of the data upload to Pride, including the Skyline file that the authors mention. That is an important resource as part of this manuscript and potentially of great interest for other scientists.

We now include the following text in the revised manuscript (underlined):

“The mass spectrometry proteomics data including the spectral library have been deposited to the ProteomeXchange Consortium via the PRIDE⁵⁰ partner repository with the dataset identifier PXD015912.

-- Methods, Data and code availability, Para. 1

1.11 The statistical processing and the algorithm RUV III C are very interesting – I understand it relies heavily on having peptides that do not change (negative controls). The authors mention in their case they choose yeast peptides as negative controls that do not change. Could the authors please elaborate or recommend how this should best be done for studies that are 100% human. The problem with using housekeeping proteins is also to know for sure what is housekeeping, what really does not change is sometimes difficult to assess. How was this done for the Bruderer et al study and for the Collins et al study – this would be important to detail more as this provides insights into how to apply this strategy in a broader way. How many negative control peptides do you need to get good statistics for this approach – was that assessed. As the authors are using likely many yeast peptides this seems to be working fine – how many negative control peptides do the authors use and what is the minimal that does still work ?

In this study, we used $n = 1,622$ yeast peptides as negative controls. The number of negative control peptides is important, and the improvement in correlation coefficient is less pronounced with fewer peptides. For example, $n = 500$ negative control yeast peptides confers a median correlation coefficient of 0.29, while $n = 1000$ negative control yeast peptides confers a median correlation coefficient of 0.56. In comparison, the full set of $n = 1,622$ yeast peptides confers a median correlation coefficient of 0.62 (see *Figure 3e*). We now include further discussion regarding the selection of negative control peptides in the *Methods* of the revised manuscript, as shown below (underlined). Specifically, we provide greater insight into how researchers might select negative controls in other studies, such as those that are 100% human, and we point the reader to a few key pieces of literature where they will find in-depth discussion of the queries raised by the Reviewer.

“Negative controls are peptides whose measurements are known, or expected to be, unaffected by biological variation in the data (that is, they are not part of the biological signal) but will still be affected by unwanted variation. There are no pre-defined rules regarding the choice of negative controls, and domain-specific knowledge and human judgement are most important^{28,42,43}. Endogenous genes will generally be a better choice of negative controls than will spike-ins, because the former reflect more unwanted variation in the experiment. For detailed discussion regarding how best to choose negative controls, and the optimal numbers of negative controls to use, readers should refer to References^{28,42,43}.”

-- Methods, Normalisation to remove unwanted variation, Para. 2

For the Bruderer et al and Collins et al studies, see below (modifications made in the revised manuscript are underlined):

Bruderer et al:

$n = 706$ negative control variables were selected by using transitions that differed most between batches as measured by a two-sided unpaired t -test.

-- Supplementary Note, Para. 2

Collins et al:

“For normalisation by RUV-III and RUV-III-C, peptide intensity data were log-transformed and of the 8,555 human peptides that were detected in every sample, the first 500 peptides were held out to assess the quality of the normalisation and the remaining 8,055 were used as negative control peptides for normalisation.”

-- Methods, Multi-laboratory study data analysis, Para. 2

1.12 It would be important to know how this method RUV-III-C works overall for low abundant peptides – is that more challenging?

The improvement in correlation coefficient after application of RUV-III-C is most pronounced for high-intensity peptides but is still evident in low-intensity peptides (see *Revision Figure 3*). Lower intensity peptides are observed in fewer samples than higher intensity peptides (see *Revision Figure 1* in Response 1.7), meaning that there are fewer data points available from which to estimate and remove unwanted variation. These findings are summarised below, whereby we selected the highest and lowest intensity human peptides ($n = 1,000$) in Sample 6 (containing 100% human tissue) and measured the correlation with the human tissue dilution series across the experimental dataset.

Revision Figure 3 – Correlation coefficients from Pearson correlation of $n = 1000$ of the highest abundant (left) and lowest abundant (right) human peptides with human tissue dilution proportions. Median Pearson correlation (r) and R^2 from each distribution are shown in italicised blue text. A black dashed line indicates the median correlation coefficient before normalisation. Correlations were calculated using \log_2 -transformed peptide intensities and ovarian cancer tissue proportions.

1.13 The technical replacement strategy is that dependent on the retention time stability. I know the authors use iRT standards – it would be worth to discuss this a bit more.

We did not find that iRT standards or consideration of retention time stability were required for satisfactory results relating to technical replacement. Retention time stability is important for peptide detection and alignment between multiple samples (see Parker et al *Mol Cell*

Proteomics 2015), whereas the technical replacement strategy only requires assignment of replicates after peptide quantification.

1.14 Overall the authors focus mainly on the MS metrics and performance – was there an overall effect of chromatography performance?

Our experimental design did not enable us to fully differentiate between performance degradation of the mass spectrometer and liquid chromatography over the course of the experiment. However, in most cases we found that instrument performance improved when the mass spectrometer underwent cleaning (see *Figure 3a*). Further, the minimum retention time alignment threshold implemented by OpenSWATH ensures that only files with sufficient correlation of iRT peptides are processed. For these reasons, we believe that the mass spectrometer was most likely to be the factor driving the observed performance degradation. However, we add the following text (underlined) into the *Discussion* of the revised manuscript to make this limitation clear to the reader:

“Our results have other practical implications for the design of large-scale studies of complex samples such as cancer tissues. Our experimental design did not enable the effects of MS and liquid chromatography performance to be fully distinguished, and performance benefits may be achieved by varying the existing maintenance schedules of each of these pieces of equipment.”

-- *Discussion, Para. 4*

1.15 Acquiring 3 technical replicates per sample is often rather challenging for laboratories and many laboratories do not have as many instruments available. Typically, as there is so much more variability in biological samples it has been considered having more biological replicates is beneficial. I do think this is a challenge of this particular design for many labs to implement. If one does have enough biological replicates would one not rather acquire data for the biological replicates instead of acquiring 3 technical replicates for each biological sample? I can rather see the point with small clinical cohorts as the authors point out.

The relative merits of biological versus technical replicates in an experiment will be context-specific (see text below). However, if a researcher chooses that biological replication is more valuable to their experimental design, *ProNorM* can still be applied to improve the reproducibility of their experimental data. While technical replacement requires that every sample be measured in technical triplicate, RUV-III-C does not. RUV-III-C only requires that enough samples in a cohort are replicated in order to adequately measure and subtract the unwanted variation. We have modified the *Discussion* in the revised manuscript (see below, underlined) to address this point:

“Increasing technical replicate MS runs clearly comes at a cost. Increasing the number of technical replicates may be a useful strategy for analysis of small clinical cohorts, such as those available from rare diseases or clinical trials, or where researchers would derive benefit from a data matrix with fewer missing values. In the absence of technical

triplicates for all samples, RUV-III-C can still be applied to a cohort independent of technical replacement, provided that at least some samples are replicated at various points during the experimental period.”

-- Discussion, Para. 3

1.16 Just reporting CV's as measure for reproducibility should be extended to also do some measurements of true quantification - forming ratios and reporting those. In this study design this is possible as different amounts of human material is in each individual sample. Figure 5 mentions some of that but rather than just reporting incidence of significance. Could the authors form some ratios at different abundance levels and see how those change or maintain significance. What is the robustness of quantification which is often much more important than a CV.

We found it challenging to identify entirely convincing ovarian-cancer specific peptides. Most human peptides are not tissue-specific, but rather are present at varying concentrations in different human tissues. We have made some attempts to identify ovarian cancer tissue-specific peptides (see *Figure 2c*), but we are careful only to use these peptides to demonstrate a linear relationship rather than calculated fold changes. This is because ratios of peptides that are not perfectly specific to ovarian cancer tissue will be miscalculated if they are present even in low concentrations within the constant prostate cancer tissue background. (We do report ratios of yeast peptides before normalisation in *Supplementary Figure 2g* of the original manuscript. However, it is not appropriate to analyse yeast peptides after normalisation, as we used these peptides for the purposes of normalisation itself).

We agree that CVs do not enable researchers to assess the impact of a normalisation approach on biological variation and we hope to have highlighted this for the reader in our manuscript: *“it is possible for normalisation to obscure biological changes when removing technical variation.”* To quantitatively measure the effects of normalisation beyond CVs, we draw attention to our analysis in *Figures 3b,d-e* in the original manuscript, where we demonstrated that *“the median Pearson correlation across each peptide increased from 0.25 before normalisation to 0.62 after normalisation, consistent with an improvement in specific signal detection”*. We believe that this is the key measure of quantitative accuracy after normalisation, and is preferable to ratios for our experimental design.

Reviewer #2

2.1 The overall results are quite expected, particularly in light of the fact that conceptually similar work has already been published (Collins et al, Mol Cell Proteomics 2017) which asks the question why it should be published in Nature Communications rather than in a more specialized journal.

We note that Collins et al was published in *Nature Communications* (not *Molecular Cellular Proteomics*). Collins et al assessed DIA-MS reproducibility during a one-week period in multiple laboratories for $n = 229$ non-tumour samples. Our submitted work not only addresses large-scale and longitudinal DIA-MS reproducibility, but also proposes novel strategies for maximising data reproducibility. It is substantially larger in size, assessing DIA-MS reproducibility for processing of 1,560 samples. Our samples primarily comprise human cancer tissue. Our data are acquired over an extended period of time (4 months) for multiple instruments ($n = 6$), interspersed with $\sim 5,000$ other runs. It is therefore also suitable for publication in *Nature Communications*.

Specific points:

Study design and data acquisition:

2.2 The study design and Figure 1 is somewhat confusing. The sample numbering in panel 1a does not match the one in panel 1b.

The numbering in *Figure 1b* was intended to convey the run order of samples. However, we agree with the Reviewer that this apparent inconsistency in numbering may be confusing to readers. We have modified *Figure 1b* to better distinguish between the run order and the sample number to improve clarity, as shown below:

Sample:

- S1 - 0% Ovary / 50% Prostate / 50% Yeast
- S2 - 3.125% Ovary / 50% Prostate / 46.875% Yeast
- S3 - 6.25% Ovary / 50% Prostate / 43.75% Yeast
- S4 - 12.5% Ovary / 50% Prostate / 37.5% Yeast
- S5 - 25% Ovary / 50% Prostate / 25% Yeast
- S6 - 50% Ovary / 50% Prostate / 0% Yeast
- S7 - 50% Ovary / 0% Prostate / 50% Yeast
- S8 - 100% HEK293T control cell line

Figure 1. Study design. ... (b) Twenty mass spectrometry runs during thirteen 48-hour periods on each of six instruments. Each run is represented by a coloured panel corresponding to each of the eight samples (labelled S1-S8), with the run order indicated in the upper left corner. Four samples were run in duplicate and four in triplicate during each 48-hour period.

2.3 That aside, it is unclear why samples 1, 2, 7 and 8 were run in duplicate while samples 3, 4, 5 and 6 were run in triplicate (here panel 1a and 1b contradict each other regarding sample 2). Is there any particular reason for this other than this creates 20 samples? The use of triplicates throughout would have allowed for a better intra-batch assessment of reproducibility.

The sample set size of 20 was chosen to fit comfortably within a 48-hour period, with sufficient flexibility in case of instrument maintenance or unexpected delays.

We chose to run samples 2, 3, 4, and 5 in triplicate as these samples comprise the core of the dilution series (3.125%, 6.25%, 12.5% and 25% ovarian cancer tissue) – increasing the numbers of replicates of these samples would be most beneficial for the experiment. The remaining samples (1, 6, 7 and 8) consist of comparatively simpler mixtures, with each comprising no less than 50% of a given tissue/cell type. For these samples, we determined that duplicates would be sufficient. We now add the following to the revised manuscript to clarify this rationale:

“Samples were run either in triplicate (the core of the dilution series) or in duplicate, in a defined sequence comprising sets of 20 samples (Fig. 1b).”

-- Results, Study design and data acquisition, Para. 1

2.4 In addition, it is not obvious why the scheduling of the QC was chosen as it was. Four replicates were performed at the beginning and the end of the study but the times in between were different. Did the authors choose this schedule on purpose? The main text states that “Mass spectrometer maintenance schedules varied according to each individual instrument’s performance”. Does that mean that Fig 1c is just a schematic representation and the actual QC runs happened at different schedules for different instruments?

We chose the schedule for running samples before commencing the experiment. The particular design follows a distinct pattern:

- A sample set (20 samples) was run every 48 hours for one week
- A sample set was run every week for the first month
- A sample set was run every month for three months
- In the final week, the first experimental week was repeated again.

This pattern was described in the original manuscript, as shown:

“The 20-sample set was run on each instrument 13 times at spaced intervals over a four-month period (Fig 1c), where each data collection occurred over 48 hours. At study

commencement, the 20-sample set was run four times in succession (days 1, 3, 5 and 7), then once per week for the remainder of the month (commencing on days 14, 21 and 28), and then once per month for the remainder of the first three months (commencing on days 56 and 84) (**Fig. 1c**). After each instrument underwent a major clean, sample sets were run a further four times in succession (commencing on days 101, 103, 105 and 107) (**Fig 1c**).”

-- Results, Study design and data acquisition, Para. 1

The Reviewer’s interpretation of *Figure 1c* is correct. This figure is a schematic representation of what was intended for each instrument. The actual times at which instrument maintenance was undertaken vary for each instrument. Instrument maintenance during this period was performed on an as-needed basis (i.e., depending on the impact on instrument performance had by the different samples run on the instruments on the intervening days between data collection for the purposes of this study). This reflects the real-world operation of a large-scale mass spectrometry facility. This is reflected in the original manuscript text, as follows:

In Main Text: “Mass spectrometer maintenance schedules varied according to each individual instrument’s performance”.

-- Results, Study design and data acquisition, Para. 1

In Figure 1c: “Instrument maintenance as required”.

To make this clearer for the reader, we have modified the *Figure 1c* legend in the revised manuscript, as shown (underlined):

“**Figure 1. Study design.** ... (c) Mass spectrometer scheduling. Days on which 48-hour periods of data collection commenced are indicated with a black bar, and the intended instrument cleaning schedule is also indicated.”

Note that the actual days of maintenance for each instrument are shown in *Supplementary Figure 1b*. We have further modified the revised manuscript to make this clearer, as shown (underlined):

“Actual days of instrument maintenance and a few minor deviations from experimental design are shown in **Supplementary Fig. 1b.**”

-- Results, Study design and data acquisition, Para. 1

2.5 What do the authors mean by ‘base line controls’? Please clarify.

Detail about quality controls is provided in the *Methods*. Some basic descriptive information has now also been added to the *Main Text* of the revised manuscript, as shown (underlined):

“After removing files ($n = 26$) that did not pass baseline quality controls such as total numbers of proteins identified and correlation with replicate samples (see **Methods**), and discarding a small number of peptides that were inconsistently observed across the cohort ($n = 934$; see **Methods**), 1,527 runs were retained (**Supplementary Table 1, 2**). These runs comprised data from a total of 17,054 peptides derived from 2,796 proteins, of”

which 2,363 were human proteins.”

-- Results, Study design and data acquisition, Para. 2

2.6 According to the main text, 80% of proteins were supported by evidence from at least 2 peptides. This seems unusual for a DIA measurement (nowadays, this figure is even reached by DDA) unless proteome coverage was very deep. But this does not appear to be the case. Any idea why that is?

This particular ratio relies on many factors, including the original complexity of the tissue proteomes extracted by pressure cycling technology and the underlying features of the library. The spectral library used in this study comprised approximately 70% of proteins with support from two or more peptides. For this reason, the reported value of 80% of proteins quantified with two or more peptides does not seem unusual. The majority of proteins quantified in our study were supported by two or more peptides. To clarify this, we now provide additional information in *Supplementary Figure 2a* of the revised manuscript so that the reader can find both individual and cumulative statistics regarding peptide support for each protein, as shown:

Supplementary Figure 2. Distribution of non-normalised SWATH-MS data. (a) Cumulative (blue; left axis) and actual (red; right axis) numbers of peptides supporting each protein identification. Blue dotted lines indicate the point at which the cumulative number of proteins have support from at least two peptides.

2.7 It would be useful to state in the main text roughly how many proteins/peptides were identified in the QC runs (say on average in each and collectively in all and how many were shared between all QC runs of the same type) so that readers get a better idea about how many data points went into the analysis that follows.

The following text (underlined) is now included the revised manuscript:

“Many peptides were consistently identified in technical replicates (Supplementary Fig. 2c) and peptide intensities were broadly consistent across the six instruments and the eight samples (Supplementary Fig. 2d, e). Of 13,485 likely true-positive peptides (i.e., peptides observed in > 10% of n = 151 replicates of Sample 8, derived from HEK293T

cells), n = 10,109 peptides (75%) were observed in at least 75% of the replicates.”
-- Results, Study design and data acquisition, Para. 2

Baseline data reproducibility during a single experimental week:

2.8 For the reproducibility analysis of the short term QC samples, why was only the second set of samples used? Does the first set of samples show the same results?

Our experimental design specified that where possible, days 101, 103, 105 and 107 should occur after each instrument underwent a *major* clean. The relationship of the first week of data collection (days 1, 3, 5, 7) to a prior major instrument clean was more variable, so the second set of samples was preferentially used for reproducibility analysis. However, results reported using data acquired only on days 1, 3, 5 and 7, are highly comparable, as shown in *Revision Figure 4*.

Revision Figure 4. Baseline DIA-MS data reproducibility. Data shown in all plots were acquired during the first experimental week (days 1, 3, 5 and 7) and were not normalised. **(a)** Principal component analysis of \log_2 -transformed experimental data, with data points coloured by sample (left) and instrument (right). Missing values were filled with zeros. **(b)** Coefficient of variation (CV) per instrument in the HEK293T control cell line [Sample 8]. CV was calculated using frequently-observed peptides ($n = 2,950$). A black dashed line marks a CV of 20% for reference. **(c)** Mean \log_2 -transformed intensities per sample of ovarian cancer-tissue specific peptides (upper) and peptides from yeast proteins (lower), coloured by instrument. The mean peptide intensity from each sample was adjusted so that relative intensities are comparable, by dividing each value by the overall mean peptide intensity. Ovarian cancer tissue and yeast proportions are plotted on the \log_2 -scale. **(d, e)** Intensities of all peptides identified from the **(d)** prostate-specific antigen (PSA) protein encoded by *KLK3* and **(e)** the housekeeping protein encoded by *TARDBP*. Boxplots show peptide intensity, with bar plots indicating the proportion of replicate samples in which each peptide was observed. Plots are coloured according to sample, using colour-codes as shown in **(a)**.

2.9 Fig. 2a: can you comment on what the ‘streaks’ in the PCA plots are?

The streaks in the PCA plot corresponding to Principal Component 2 primarily represent the number of zero values present in each sample (see *Revision Figure 5*). As noted in the legend of *Figure 2a*, we fill missing values with zero for the purposes of calculating and plotting this figure. Some of these missing values will represent true zero values (i.e., true negative), where a peptide is not present in the mixture, or the peptide is not present above the limit of detection. In other cases, these missing values will be false negatives in which features were not identified despite ions being present at detectable concentrations.

Revision Figure 5 – Principal component analysis of \log_2 -transformed experimental data, with data points coloured by the number of zeros. Missing values were filled with zeros.

2.10 Do you already see that the performance within this short period of time shows a trend?

No, we do not see a significant performance trend over the short period of time analysed in the final experimental week (see *Revision Figure 6*):

Revision Figure 6 – Intensities of human peptides before normalisation. Boxplots are coloured by instrument, within which data are ordered from earliest experimental day (left) to latest experimental day (right) within an experimental week after cleaning (Days 101, 103, 105 and 107). Data are shown for replicates of Sample 5.

2.11 The statement that “We observed an approximately linear relationship between the intensities of ovarian cancer tissue-specific peptides” is clearly not warranted. Fig 2c has no y-axis annotation but it is clear that the data follows a (shallow) exponential loss of intensity over the ‘dilution series’. This is very surprising as the ‘dilution’ only spans a factor of 16. DIA is supposed to be very quantitative but this does not appear to be the case for this particular data.

We believe that the shallow exponential loss of intensity in this figure was attributable to the data-processing method that we applied for adjusting peptide intensities in order for their concentrations to be comparable in peptides spanning large intensity ranges. In *Figure 2c*, the concentration of each peptide is adjusted by dividing by its overall mean intensity in the experiment. In the original manuscript, we performed \log_2 -transformation before dividing by the overall mean intensity. In the revised manuscript, we now adjust the raw peptide intensity before performing \log_2 -transformation. Adopting this revised approach, we find that the apparent loss of intensity in ovarian peptides over the dilution series is no longer observed. (Irregularity in yeast peptide intensity curves over small concentration changes are likely attributed to experimental rather than measurement variation). We include this revised *Figure 2c* (shown below) in the revised manuscript. Y-axis annotation is now provided.

Figure 2. Baseline DIA-MS data reproducibility. ... (c) Relative log₂-transformed intensities per sample of ovarian cancer-tissue specific peptides (upper) and peptides from yeast proteins (lower), coloured by instrument. The mean peptide intensity from each sample was adjusted so that relative intensities are comparable, by dividing each value by the overall mean peptide intensity measured on a given instrument during the period. Ovarian cancer tissue and yeast proportions are plotted on the log₂-scale.

2.12 How do the authors explain the detection of PSA in sample 7 (no prostate) and the very large variability of the intensity of the four different PSA peptides? Is this carry-over? In addition, even though one can see an association of the intensity of TAR DNA-binding protein 43 and the proportion of the amount of ovarian tissue in the sample, the relationship is not linear. Does this perhaps imply that the QTOF detector already saturates at log₂ intensities of 13 or higher? This effect should be analysed more systematically. There are thousands of peptides for which this could be done. At this stage, it is preliminary to conclude that DIA-MS is generally reproducible.

The PSA peptides in Sample 7 (containing no prostate cancer tissue) are most likely to be false positive discoveries. This is because (a) these peptides were found at lower intensities than in other samples and (b) these peptides were observed in far fewer samples. We have modified the revised manuscript to note this observation, as shown (underlined):

“To investigate consistent detection of the basal tissue matrix, we examined prostate-specific antigen (PSA; the protein used in prostate cancer screening²⁶), and found similar intensities among PSA peptides in Samples 1-6 containing 50% prostate cancer tissue,

along with some likely false positive observations in Sample 7 containing no prostate tissue (Fig. 2d)."

-- Results, Baseline data reproducibility during a single experimental week, Para. 1

Figures 2d and 2e were generated using data that were not normalised and included measurements derived from six different instruments. Therefore, some level of variability should be expected in these measurements. After normalisation, the variability in PSA peptide intensities is far lower, and the dilution effect in peptides from the TAR DNA-binding protein 43 more closely matches expectations (see Revision Figure 7):

Revision Figure 7 - Intensities after normalisation of all peptides identified from the prostate-specific antigen (PSA) protein encoded by KLK3 (upper) and the housekeeping protein encoded by TARDBP (lower). Boxplots show peptide intensity, with bar plots indicating the number of replicate samples in which each peptide was observed.

Importantly, Figure 2 represents initial qualitative analysis that serves to orientate the reader to the dataset. We perform the systematic analyses requested by the Reviewer subsequently in the manuscript. For example, in Figure 3e we provide correlations with the human tissue

dilution series for 2,904 peptides. To make this clearer to the readers, we have modified the following text in the revised manuscript as shown (underlined):

“From these initial qualitative analyses, we conclude that DIA-MS data obtained over a short time-period are generally reproducible and can support discriminative accuracy between human tissues prior to normalisation.”

-- Results, Baseline data reproducibility during a single experimental week, Para. 1

Decrease in instrument sensitivity over time since cleaning:

2.13 Fig. 2a and 2b show a ~4x variation of peptide intensities over just 4 weeks and the authors do point this out. Although this reviewer does not question to general validity of the statements made around Fig. 2b, I would be very careful when interpreting results of Pearson correlations that only have 4 data points (why was the 50ug sample not included?) ... Again, one should be very careful not to over-interpret Pearson correlations on four data points as shown in Fig. 3d.

Please note that we assume the Reviewer is intending to refer to *Figure 3* rather than *Figure 2*, and we respond accordingly.

These Pearson correlations relate to datapoints within one of four categories, but they are calculated on many more than four data points. Importantly, variance of these data points around the mean is reflected in the calculation of Pearson correlation. Moreover, Pearson correlation is only one of the metrics used for assessing the success of our normalisation approach. The others are CV (*Figure 3f*) and significant differences in peptide intensities (*Figures 5a, 5b* and *Supplementary Figures 5a, 5b*). To address this point, we have modified the revised manuscript as shown (underlined):

“The effectiveness of RUV-III-C in normalizing replicate data was assessed with a variety of metrics, such as median experiment-wide CV of frequently-observed human peptides which decreased from 37% to 13%, and dilution linearity which improved from 0.25 to 0.67 Pearson correlation (see also Fig. 5 and Supplementary Fig. 5).”

-- Discussion, Para. 2

Regarding the “50ug sample” queried by the Reviewer, two samples in our cohort contained 50% ovarian cancer tissue (Samples 6 and 7) but neither of these samples is suitable for the correlation analyses performed in *Figure 3b*. Sample 7 contained no prostate cancer tissue and therefore the intensity of human peptides in this sample should not follow a linear relationship when compared with Samples 2-5. Sample 6 was discarded prior to normalisation as it did not contain the yeast peptides that were necessary negative controls.

Development of a method for data normalization:

2.14 It is not clear to this reviewer why the ‘novel’ normalization procedure was necessary or if it really improved the results. Looking at Fig 3c and the supplementary figures, it at least visually looked as if median centering the data

produced better intensity normalized results than RUV-III-C. At least when using all peptides for normalization. Why do the authors show the normalization based on the iRT peptides only in the main figure?

Median normalisation works by setting the overall median for each sample to the same value across a cohort. Hence, it is entirely unsurprising that the results of this procedure produce the *best-looking* normalisation results in *Supplementary Figure 3c*. However, as stated in our original manuscript, “*it is possible for normalisation to obscure biological changes when removing technical variation. This is a particular risk when cohorts comprise complex samples such as heterogeneous human tissues.*” For this reason, in our original manuscript we sought to also examine (a) iRT peptides (see *Figures 3a* and *3c*) and (b) correlations between human peptide intensity and human tissue concentration (see *Figures 3b, d-e*). These metrics both demonstrate the superiority of RUV-III-C when compared with median normalisation. To make clearer that we do not intend for the reader to conclude from *Supplementary Figure 3c* that median normalisation was successful, we have modified the related text in the revised manuscript as shown below (underlined):

“Median normalisation and median normalisation plus ComBat²⁹ for batch effect removal at first appeared to successfully normalise the entire set of peptides (Supplementary Fig. 3c), but importantly, variability remained in the intensities of individual peptides (Supplementary Fig. 3d). Further, correlation with the dilution series was only minimally improved (0.32 and 0.40, respectively; Fig. 3e).”
-- Results, Development of a method for data normalisation, Para. 3

The variation that we observed in iRT peptides indicated that subsets of peptides are more informative for determining the success of each normalisation approach than the entire set of peptides. For this reason, we included these in the main *Figure 3*, and the plots of all peptides are shown in the *Supplementary Figure 3*.

2.15 What is quite puzzling is that the new normalization procedure did not really improve results for an individual instrument (Fig. 3f). Hence, the improvements across the entire data set is surprising in its extent.

The single instrument results in *Figure 3f* in the original manuscript reflect a single week of data collection. There is very little to normalise when data relate only to a single operating week on a single instrument (see *Revision Figure 6*), so the impact of RUV-III-C in these circumstances is limited. Importantly, when data are combined across all instruments in a single week, or across the entire experimental period, the reduction in CV is much more pronounced. This is the significant contribution of our approach, and we have modified the text to highlight this in the revised manuscript as shown (underlined):

“Most importantly, our unique study design enabled insights into the sources of technical variation in high-throughput MS, leading to the development of the ProNorM methodology that can mitigate these effects particularly over long periods and/or across multiple instruments.”
-- Discussion, Para. 1

To further address the Reviewer’s comment, we have modified *Figure 3f* in the revised manuscript. The revised figure (see below) now shows CVs for each sample type across the experiment averaged over six instruments, which should be more informative for readers.

Figure 3. Peptide intensity variation during the experimental period and normalisation approaches. ... (f) Coefficient of variation (CV) of frequently-observed human peptide intensities, calculated for each sample during the experimental period across all instruments. A black dashed line marks a CV of 15% for reference.

Technical replacement of missing values:

2.16 Missing value imputation is a tricky business and the authors may want to point this out. The Figure 4a is a good way of showing this but I would suggest to add a line that shows the intensity distribution of the high-confidence peptides to be able to see if MCAR peptides are similar or dissimilar to the frequently detected peptides. However, I don’t think it is necessary to show six almost identical plots to make this point. One could show one or two and move the rest to the supplement.

We have incorporated the following text into our revised manuscript (underlined):

“The numbers of peptides observed across replicates varied up to three-fold during the study, with highest peptide numbers recorded after instrument cleaning (Fig. 4b). Adequately dealing with missing values in a large dataset can be challenging. We developed a new method termed technical replacement that leverages technical replicates at different phases of instrument maintenance, to replace missing values with plausible intensities.”

-- Results, Technical replacement of missing values, Para. 2

Please note that we use the term ‘replacement’ in our manuscript to differentiate our method from traditional ‘imputation’ methods that use model-based estimates generated *in silico*.

Peptides defined as *likely missing completely at random (MCAR)* are in fact frequently detected peptides. *MCAR* peptides were defined as such if they were missing in an average of \leq one of six replicates, across random groups of six replicates. *Figure 4a* demonstrates that *MCAR* peptides were generally present at high intensities in our cohort.

We have moved some of the panels from *Figure 4a* to *Supplementary Figure 4*, as shown:

Revised Figure 4a

Figure 4. Missing values and results from technical replacement. (a) Distribution of median non-missing intensity of each peptide designated as likely missing completely at random (MCAR) and missing not at random (MNAR) in Samples 1 and 2.

Revised Supplementary Figure 4a

Supplementary Figure 4. Missing values and peptide identifications after technical replacement. (a) Distribution of median non-missing intensity of each peptide designated as likely missing completely at random (MCAR) and missing not at random (MNAR) in Samples 3, 4, 5 and 7.

2.17 Similarly, it would suffice to show one or two instrument panels in Fig 4b.

We agree with the Reviewer that all six instrument panels are not required for adequate interpretation of *Figure 4b*. However, we feel that as currently shown, *Figure 4b* maintains an important consistency with *Figure 3a*.

2.18 I suggest that the extra space generated in this way could be used to illustrate that/how imputing missing values improves reproducibility. This is not really captured in Fig.4c but would be important to show. The current Fig 4c raises the question as to how important the missing value imputation actually is. Even without any ‘technical replacement’, the true positive ISs are already at >97% (at 5% FP). Increasing the true positives by just 2% comes at the cost of increasing false positives by almost a factor of two. The authors should therefore tone down the importance of this imputation (albeit acknowledging that this does not seem to hurt much).

We have reduced the size of *Figure 4c* and incorporated a new figure (*Figure 4d*) in the revised manuscript. *Figure 4d* demonstrates the manner in which technical replacement improves reproducibility, as described below in the revised text (underlined):

“When technical replacement was applied across replicates acquired on three instruments (where missing values were only replaced when a peptide was observed in two replicates), ~20% of missing values could be replaced with plausible non-zero intensities. Technical replacement became more effective as samples comprised increasing proportions of ovarian cancer tissue (Fig. 4d), likely because higher peptide intensities led to more reliable detection of each peptide in at least two replicates.”

-- Results, Technical replacement of missing values, Para. 2

Figure 4. Missing values and results from technical replacement. ... (d) Proportion of missing values replaced in each sample after technical replacement. Data are shown for triplicates with missing values replaced when a peptide was observed in two of three replicates, i.e., 3MS (≥ 2 MS).

Figure 4c is intended to show the true and false positive rates resulting after different methods of technical replacement. The reader should then infer from this figure which method is most suitable. We have modified the text below (underlined) to make this clearer in the revised manuscript:

*“We defined likely true and false positives (see **Methods**) and, after technical replacement, the likely true positive rate increased from 97.2% to over 99% (**Fig. 4c**). With no constraints on replacement, the likely false positive rate increased with each subsequent technical replicate, from 4.8% to 24.6% with six replicates (**Fig. 4c**). Hence, to constrain false positive accumulation, missing values are best replaced only when a peptide is observed in more than one replicate. Only in this manner can the true positive rate be increased without considerably impacting upon the false positive rate.”*

-- Results, Technical replacement of missing values, Para. 2

To satisfy the Reviewer’s comment regarding the importance of technical replacement, we have modified the related text in the revised manuscript to provide a more nuanced discussion of when technical replacement might provide benefit, as shown (underlined):

“Increasing technical replicate MS runs clearly comes at a cost. Increasing the number of technical replicates may be a useful strategy for analysis of small clinical cohorts, such as those available from rare diseases or clinical trials, or where researchers would derive benefit from a data matrix with fewer missing values. In the absence of technical triplicates for all samples, RUV-III-C can still be applied to a cohort independent of technical replacement, provided that at least some samples are replicated at various points during the experimental period.”

-- Discussion, Para. 3

Simulation of cohort analyses in proteomics:

2.19 From reading the main text, it is not very clear what Fig. 5a shows.

We now incorporate the following additional text into the revised manuscript to improve clarity for the reader regarding Figure 5a (underlined):

*“Examining frequently-observed human peptides and averaging across technical triplicates, we found that ProNorM conferred a vast improvement on our ability to detect and quantify significant differences in human peptide intensities between samples containing different proportions of human tissue ($P < 0.0001$ by unpaired *t*-test; **Fig. 5a**).”*

-- Results, Simulation of cohort analyses in proteomics, Para. 1

2.20 The same applies to Fig. 5b. While the reviewer can guess what the solid blue lines show, the grey areas and the lines in them are not clear. Perhaps I do not get the figure at all but the data in 25% ovarian cancer tissue proportion panel in Fig 5b implies that it was impossible to distinguish samples with 25% or 50% ovarian

cancer content. This would be highly unexpected as this should be the easiest of all comparisons. Please clarify.

We have simplified *Figure 5b* in the revised manuscript. We have also added additional labelling and detail in the legend. We now also provide *Supplementary Figure 5b*, to allow for further information to be shown. Please see the revised plots below.

Figure 5. Simulation of cohort analyses for discovery proteomics. ... (b) Percentage of frequently-observed human peptides that were significantly different (vertical axis) in simulated cohorts of varying sizes (horizontal axis). Plots show comparison between Sample 2 (containing 3.125% ovarian cancer tissue) and Samples 2-5 (containing 3.125% to 25% ovarian cancer tissue), without normalisation (left) and after ProNorM (right). Shading denotes 95% confidence intervals derived from ten iterations of random selections of replicates of each sample.

Supplementary Figure 5. Simulation of cohort analyses for discovery proteomics across normalisation methods. ... (b) Percentage of frequently-observed human peptides that were significantly different (vertical axis) in simulated cohorts of varying sizes (horizontal axis). Plots show comparison between Sample 4 (containing 12.5% ovarian cancer tissue) and Samples 4-5 (containing 12.5% and 25% ovarian cancer tissue), without normalisation (left) and after ProNorM (right). Shading denotes 95% confidence intervals derived from ten iterations of random selections of replicates of each sample.

Prediction of tissue proportion in a complex mixture:

2.21 It would indeed be exciting if this approach could work. However, the machine learning part is incomprehensible from reading the main text and, more importantly, how relevant is it to train such a classifier based on a mixture of two cancer types and yeast that are mixed into one sample. Such a situation would never occur when analysing clinical specimen. Furthermore, in light of the data from the previous section, this reviewer seriously doubts that that any such predictions can be meaningfully performed on the very limited amount of available data used for training. How do the authors make sure that the model is not entirely overtrained? In my opinion, a 10-fold cross validation does little/anything to validate the predictions. Instead, another data set should have been generated to test the performance of the trained classifier.

The intention of the analysis described in *Figure 6* is to demonstrate that DIA-MS data are suitable for creating a model that can accurately predict the underlying composition of a peptide mixture. When analysing clinical specimens, we can use the same strategy to re-train a model that is based on the clinical cohort of interest. The results reported in our current study suggest that with appropriate training data, data obtained by DIA-MS could be successfully used to develop a model that could predict, for example, the proportion of cancer cells in a tissue biopsy in a real-world clinical scenario.

Our machine learning predictions were made using only human peptides (not human and yeast peptides) from different cell types, making our simulated scenario more similar to what might take place in a real-world clinical setting. We have modified the text in the revised manuscript to clarify this for the reader (see below, underlined).

Regarding the Reviewer's comments suggesting limited data and over-training, we do not believe that these are valid criticisms of our model. We applied an MLP regressor which is regularized – a technique to mitigate overfitting/over-training. Furthermore, we developed four separate models, each trained on three concentrations of ovarian cancer tissue and tested on the remaining concentration, which was entirely unseen by the regressor. For example, to assess the prediction performance of the model for 3.125% ovarian cancer tissue concentration ($n = 225$ samples \times 13,721 peptides), the regressor was trained using data only from samples containing 6.25%, 12.5% and 25% ovarian cancer tissue ($n = 695$ samples \times 13,721 peptides). Thus, the reported errors were out-of-sample or generalisation errors. We also note the large numbers of data points for both training and testing, which should be a

sufficient sample sizes for this analysis. Ten-fold cross-validation was only employed for parameter tuning, as described in the original manuscript.

We have further modified the text (underlined) to improve clarity for the reader, as follows:

“The ability to estimate the proportion of a specific tissue component in a complex sample could have diagnostic utility (for example, the proportion of cancer cells in a tissue biopsy)³¹. To demonstrate whether such an analysis might be possible using DIA-MS data, we applied machine learning to estimate the proportion of ovarian cancer tissue among the prostate cancer tissue background after ProNorM. We trained four regularized regression models on the entire set of human peptides measured in three concentrations (for example, 6.25%, 12.5% and 25% ovarian cancer tissue; n = 695 samples) to avoid overfitting. Hyperparameters were tuned by ten-fold cross validation, and we report out-of-sample errors after testing the models on the remaining independent concentration (for example, 3.125% ovarian cancer tissue concentration; n = 225 samples; see Methods; Fig. 6 and Supplementary Fig. 6). Our models successfully predicted ovarian cancer tissue proportions with a mean absolute error of as low as 0.8% (Fig. 6). Therefore, with appropriate training data, DIA-MS could be applied to measure tissue proportions in a complex background, demonstrating significant potential for clinical proteomics.”

-- Results, Prediction of tissue proportion in a complex mixture, Para. 1

Other points:

2.22 This reviewer suggests that the authors came up with an alternative title that better reflects the content of the paper. The current one is too general and likely turns away a lot of potential readers.

We have modified the title of the revised manuscript, which now reads:

“Strategies to enable large-scale proteomics for reproducible research”

2.23 There are a few minor issues with language and grammar that should be fixed.

We have corrected any language and grammatical errors discovered during the revision process.

2.24 The text could be clearer in places. E.g. in the abstract, it is not clear what ‘control’ is. This becomes clear in the main text (HEK293T cells). Also, the term ‘dilution series’ is not accurate for the description of samples 1-6. The total peptide content in these samples is the same but the proportions of the ovarian cancer and yeast samples are different.

We now specify in the abstract of the revised manuscript that our control was HEK293T cells. We have also clarified wherever appropriate that we intend to refer specifically to a dilution series of either ovarian cancer or human tissue.

Co-worker I

Major comments:

3.1 In particular in the first paragraphs there is a lot of repetition of what is already known. For example, the reproducibility chapter was already covered by Collins, et al 2017 in a more comprehensive way (intra-day variation, inter-day variation, even inter-lab variation). Other studies focus on the reproducibility and repeatability of DDA data (Tabb et al, 2010). Another example is the sensitivity decrease post cleaning chapter. The results are common knowledge.

The *Introduction* is written to give the reader background knowledge relating to the field. This allows our manuscript to be clearly placed within the existing literature.

3.2 It would be more desirable if the entire data set is analysed in terms of long-term reproducibility which is novel and interesting! In this case it would be helpful to name the factors that mostly influence reproducibility.

The main focus of this manuscript is reproducibility over-time and we have identified instrument maintenance as the key contributing factor. This is outlined in the *Introduction* in the original manuscript, as shown:

“When measurements are acquired over short experimental periods, DIA-MS data collected across different laboratories can be adequately combined. However, to achieve datasets of sufficient size to support robust discovery, data collected from multiple instruments over long periods must be effectively integrated. It is therefore imperative that the impact of factors affecting the reproducibility of peptide quantitation are known, and that data analysis techniques are developed to optimise reproducibility under these circumstances.”

-- Introduction, Para. 3

To address the Co-Worker’s comment, we have modified the revised manuscript to further clarify our aims, as shown (underlined):

“To enable acquisition of reproducible large-scale data for studies of cancer and other diseases, the aim of this study was to document the degree of long-term inter-instrument and temporal variation in the discriminative proteomic profiles of cancer tissues analysed in a high-throughput facility, and to develop methods to improve reproducibility.”

-- Introduction, Para. 4

3.3 The RUV-III-C algorithm is a variant of RUV-III. As cited in the manuscript, another group already implemented this algorithm for gene expression data. In the present study, the authors transferred this knowledge to proteomics data and extended the algorithm to data sets with missing values.

This statement is correct.

3.4 One selling point of DIA is that it has in general fewer missing values than DDA. Why do the authors apply their method not to DDA data or show the difference between DDA and DIA. It would be great if the normalization procedures could be applied to all kind of proteomic data sets. This makes an even stronger point than comparing the results of two mass spectrometric platforms (Sciex vs Orbitrap), because there the output data is very similar.

The purpose of our analysis was to assess the reproducibility of large-scale DIA-MS data. For this reason, we designed this large-scale study to acquire 1,560 MS runs on six instruments across a four-month period. We agree that analysis of DDA-MS data might be of interest, but it is outside the scope of this present study. Our analysis of DIA-MS data from both SCIEX (see *Supplementary Figures 3e* and *3f*) and Orbitrap (see *Supplementary Note*) instruments importantly demonstrates the applicability of *ProNorm* beyond our specific experimental design.

3.5 How long does it take to normalize data and replace missing values with ProNorm? Which computational resources are required?

In brief, the time required to run RUV-III-C and perform technical replacement will naturally depend on the size of the dataset being analysed. Unless a dataset is extremely large, both processes can be completed on a standard laptop or desktop computer, and we do not expect computational resources to be a limiting factor. In cases where researchers do find themselves reaching the limits of their in-house computational resources, researchers can utilise cloud computing platforms to great success.

To provide a quantitative answer for the Co-Worker, we adopt *big-O* notation, which describes the rate at which computational requirements change as the input grows. For example, $O(n)$ indicates that a doubling of input size doubles the computational requirements. $O(n^2)$ indicates that a doubling of input size multiplies computational requirements by four times. For each peptide, RUV-III-C first computes an orthogonal projection, which in practice will have the complexity $O(\text{num_per_peptide}^3)$ of matrix multiplication, where *num_per_peptide* is the number of mass spectrometer runs in which the current peptide is detected. We then compute the first k singular values of a *num_per_peptide* \times *num_per_peptide* symmetric matrix. Complexity will therefore depend on k and *num_per_peptide*. Given that *num_per_peptide* will be less than the number of samples, a very conservative upper bound for the runtime of the entire algorithm will be $O(\text{num_samples}^3 \times \text{num_peptides})$. The computational resources required for technical replacement are trivial, and this method requires $O(n)$ time. As reported in our manuscript, technical replacement was implemented using Python code. Each iteration of technical replacement reported in *Figure 4c* was completed in under an hour using a standard laptop computer.

To address the comment raised by the Co-Worker, we have added the following text to the revised manuscript (underlined):

“We will execute as many runs of RUV-III on different subsets of acquisitions as there are peptides that we wish to normalise. Unless a dataset is extremely large, we do not expect computational resources to be a limiting factor.”

-- Methods, Normalisation to remove unwanted variation, Para. 4

3.6 Do the authors attempt to include their procedure as an option in standard DIA analysis software tools?

We would greatly encourage the inclusion of our methods in standard DIA analysis software tools. However, we have not developed any such tools ourselves, and we therefore do not have the option to incorporate *ProNorM* into these software tools. Once our method becomes accepted by the scientific community through publication of our manuscript, we would encourage *ProNorM* to be integrated into standard tools. In the meantime, our methods are clearly described in our manuscript, and we have made RUV-III-C available as a public package for other researchers to utilise in their own pipelines.

3.7 Is the data first normalized and then imputed or the other way around? Why was this order chosen (Karpievitch, 2012 show that the order matters)?

Data are normalised first and imputed second. By doing so, we ensure that any datapoints transferred during technical replacement are first made to be comparable across replicates and instruments.

3.8 It would be great if authors could show a biological example where this normalization and imputation strategy outperforms standard normalization and imputation strategies. Maybe there are examples where without these novel algorithms, no biological conclusion can be drawn.

It is challenging to obtain public DIA-MS datasets (a) of sufficiently large-scale and (b) in which a clear biological endpoint can be determined that is not yet known. Many existing large-scale studies were conducted for the purposes of exploratory research, and unknown biological distinctions are not trivial to discover. Most importantly, conclusions about the validity of a novel normalisation method cannot be drawn unless the underlying biology is known. *Figures 5 and 6* are intended to serve as a proxy for biological examples, demonstrating how *ProNorM* might enable conclusions to be drawn that would not otherwise have been possible without this algorithm. Our experiment is specifically designed to determine whether our normalisation method works as expected. Many previous studies focusing on quality-control and reproducibility also use standard samples as the most ideal reference point for assessment (for example, see Navarro et al *Nat Biotechnol* 2016).

We expect that as *ProNorM*, together with our recommendations for experimental design, are adopted by researchers conducting large-scale DIA-MS studies into the future, that our methods will enable novel biological discoveries to enter the literature.

Minor comments:

3.9 Introduction and study goal: The study goal is much better outlined in the abstract than in the introduction. Major aim was not the evaluation of reproducibility, but the development of a new algorithm that improves reproducibility.

We have modified the *Introduction* in the revised manuscript to more clearly state the aim of our study, using language similar to that used in the *Abstract*, as shown (underlined):

“To enable acquisition of reproducible large-scale data for studies of cancer and other diseases, the aim of this study was to document the degree of inter-instrument and temporal variation in the discriminative proteomic profiles of cancer tissues analysed in a high-throughput facility and to develop methods to improve reproducibility.”

-- *Introduction, Para. 4*

3.10 Page 6, paragraph 2 and methods section: ‘Many peptides were consistently identified...’ Filtering happened before. It would be also great to show how consistent the data is without filtering.

This information was provided in the *Methods* of the original manuscript. Here we described that filtering excluded only a small number of samples ($n = 26$) and peptides ($n = 934$), leaving 1527 samples and 17,054 peptides for analysis. These figures demonstrate that very minimal filtering was performed given the scale of this study, giving the reader an indication of the consistency of the dataset prior to filtering. We have modified the main text in the revised manuscript to provide further information, as shown (underlined):

*“After removing files ($n = 26$) that did not pass baseline quality controls such as total numbers of proteins identified and correlation with replicate samples (see *Methods*), and discarding a small number of peptides that were inconsistently observed across the cohort ($n = 934$; see *Methods*), 1,527 runs were retained (*Supplementary Table 1, 2*). These runs comprised data from a total of 17,054 peptides derived from 2,796 proteins, of which 2,363 were human proteins.”*

-- *Results, Study design and data acquisition, Para. 2*

3.11 Page 6, paragraph 3: Only quantitative evaluation, but what about the quality of the data, e.g. repeatability (similar to Tabb et al). What about intra-day, inter-day, inter-instrument analysis?

We intend for *Figure 2* to be only a general introduction to our dataset. Inter-instrument and qualitative analyses have been performed in detail in subsequent figures (see specifically *Figures 3* and *4*). To make the intended purpose of *Figure 2* clearer for the reader, we have modified the text of the revised manuscript as follows (underlined):

“We next investigated longitudinal variation in experimental data over the entire study period, as was the primary aim of this study.”

-- *Results, Decrease in instrument sensitivity over time since cleaning, Para. 1*

As noted by this Co-Worker in Comment 3.1, the requested elements of intra-day and inter-day analysis have been discussed by Collins et al (*Nat Commun* 2017) and Tabb et al (*J Prot Res* 2010). We do not feel that additional analysis in this area would add value to our manuscript, as the primary stated purpose of our study is to assess long-term reproducibility.

3.12 Additionally, it would be very helpful to get the acquired data points per peak so that the reader can get a feeling of the quality of quantification.

The average peak width in this study was 40.87 seconds. With a cycle time of 3.2 seconds, we therefore acquired an average of 12.8 data points per peak. We now provide this information in the *Methods* of the revised manuscript, as shown (underlined):

“MS2 spectra were collected in the range 100-2000 m/z for 30 ms in high resolution mode and the total cycle time was 3.2 seconds. The average peak width was 40.87 seconds, meaning that an average of 12.8 data points were acquired per peak.”

-- Methods, DIA-MS data acquisition, Para. 2

3.13 Page 6/7: What about chromatographic stability ◊ easy check with iRTs, does this influence the identification and quantification of peptides?

Retention times were generally very stable during the experimental period (see *Revision Figure 8*). We observed only minor changes in retention time associated with probe, analytical column or trap column changes, which occurred infrequently during the experimental period.

Revision Figure 8 – Observed retention time of indexed retention time peptides (n = 29) in Sample 5 during the experimental period. Times of instrument maintenance are indicated with coloured vertical lines and results from each instrument are shown in separate panels.

3.14 Page 7, paragraph 1 and 2: What about the abundance of the chosen peptides?

This information is provided in *Figure 3b*, where the y-axis indicates the intensity of the chosen peptide in each sample.

3.15 Page 10, paragraph 1: ‘We then replace missing values ... observed for that peptide in a technical replicate’. What happens if the replicates do not contain this peptide? This might be a peptide that is regulated and might be of interest or is a false identified one.

We only replace a missing value if it is observed (i.e., identified by the OpenSWATH pipeline) in a technical replicate. We have modified the relevant text in the revised manuscript to make this clearer for the reader, as follows (underlined):

*“We then replaced missing values with a value sampled from a normal distribution centred around the mean normalised intensity observed for that peptide if it were identified in a technical replicate (see **Methods**).”*

-- Results, Technical replacement of missing values, Para. 2

3.16 Page 11, paragraph 2: what’s the quality of the 25 replicate samples? From which time points were they taken (after cleaning, before and after cleaning, etc)? A comparison to standard normalization techniques like grand mean normalization would strengthen the conclusion. Comparison to non-normalized data and a conclusion that the algorithm improves statistical significance is obvious.

We used all samples that contained 25% ovarian cancer tissue for *Figure 5b*, regardless of time since cleaning. Please note that we have modified *Figure 5b* to simplify the contents and provide more detailed labelling and legend in the revised manuscript (see Response 2.20). To provide further explanation here, *Figure 5b* functions much like a power analysis – extending the results in *Figure 5a* and *Supplementary Figure 5a* by simulating scenarios in which we might have smaller numbers of samples in a cohort. *Supplementary Figure 5a* compares the results of *ProNorM* with data normalised using other approaches (i.e., median normalisation and median normalisation plus ComBat). This figure demonstrates that *ProNorM* conferred the greatest improvement in statistical significance when compared with data normalised by other methods, and not only when compared to non-normalised data.

3.17 Page 20, 515, 516: Precipitation of SDC, page 20: no washing

We clarify that the queried step relates to precipitation of SDC, not peptides. Hence, no washing step was required. Furthermore, samples were subjected to C18 SPE clean-up prior

to mass spectrometer analysis. We refer the Co-Worker to the *Methods* of the original manuscript, which describes these steps in detail.

3.18 Page 20, 530; Page 21, 548: Library was generated with 0.5 % FA, DIA measurements with 0.1% FA. Any reason for that?

We have corrected the relevant *Methods* text in the revised manuscript to indicate that the fractions were resuspended in 0.1% FA rather than 0.5%.

3.19 Page 21, 561: Why using three different search engines? What's the overlap between them?

Three complementary search engines were used to increase the confidence and sensitivity of our identifications when compared with single-search engine processing. The utility of this approach has been demonstrated in the literature and is commonly used in other studies (see Shteynberg et al *Mol Cell Proteomics* 2013, Nesvizhskii et al *J Proteomics* 2010 and Stewart et al *Nat Commun* 2019).

We observed significant overlap between the spectral libraries generated by each search engine when considering peptides from prostate and ovarian cancer tissue (*Revision Figure 9*). However, we found that Mascot identified far higher numbers of both total and unique PSMs from yeast than did X!Tandem and MS-GF+ (*Revision Figure 9*). This demonstrates both the importance and utility of adopting a multiple complementary search engine approach to spectral library generation, as was the case in this study.

Revision Figure 9 – Numbers of total (left) and unique (right) PSMs identified by each search engine for each sample indicated.

3.20 Page 22, 574: ‘peptides (including modified peptides): Use precursors information or use the term modified peptides instead.

We have modified the relevant text in the revised manuscript, as shown (underlined):

“The resulting library contained 31,614 precursors from 6,865 proteins, of which 5,013 are human proteins.”

-- *Methods, Spectral reference library generation, Para. 4*

3.21 Page 22: 7.000 proteins from 39.033 peptides. What is total protein/peptide number without all the filtering?

These data were provided in the *Methods* of the original manuscript. We have now added this information to the *Main Text* in the revised manuscript – see Response 3.10.

3.22 Page 23: 100 variable windows with 3.2 sec cycle time \diamond dp/peak?

See Response 3.12.

3.23 Page 24, 620: Why excluding samples with $R2 < 0.9$ in replicates? This is a study about reproducibility. Why excluding files with less than 1.200 proteins, what's the explanation?

In a large study acquiring $> 1,500$ MS runs, we naturally expect that some data will be of particularly poor quality and beyond that which is due to general instrument variation and performance degradation. As would typically be done in other experiments, we chose to remove any files that appeared to be particularly poor (i.e., outliers) before proceeding with normalisation. The limits of 0.9 Pearson correlation and 1,200 proteins were data-driven thresholds, below which were only a handful of files ($n = 26 / 1,560$). We reasoned that removing such files enabled a fairer assessment of DIA-MS reproducibility, as it would mimic the data pre-filtering process that other laboratories would typically go through when conducting a large-scale biological study. To ensure transparency, this is clearly documented in both the *Main Text* and *Methods*, and the discarded files remain available for download via PRIDE under the accession number PXD015912.

3.24 Page 27, 695: What about propagation of variation when scaling? \diamond Normalization is applied already to this set,

Any variation inherent to these yeast peptides will be present when intensities are scaled. RUV-III-C uses the variation in negative control peptides (which are otherwise expected to be unchanging across the experimental period) to estimate and then subtract unwanted variation from the remainder of the cohort. Negative control peptides are un-normalised at the time of input into RUV-III-C.

Coworker II

Key points in this manuscript

4.1 *We demonstrate for the first time the reproducibility of large-scale DIA-MS measurements over extended periods and across multiple mass spectrometers. This is true, but I personally doubt the value of this experimental design.*

This experiment presents a unique quality control assessment, reflecting how a large centre might run clinical samples on a daily basis, while ensuring statistical rigor. We believe that the results presented in this manuscript, along with the improvements made after this review cycle, will be a testament to the value of our experimental design.

4.2 *To reflect real-world operating conditions, approximately 6,500 MS runs were carried out over a four-month period, during which the mass spectrometers received minor and major maintenance on an as-needed basis. I think this is not very meaningful, unless they aim to generate dataset under different MS sensitivity at the beginning. The sensitivity loss is well known in proteomics labs. In the real-world operating conditions, researchers can easily monitor the machine performance by QC sample, the maintenance can be very flexible based on the QC data instead of an as-needed basis. In such big project, QC samples must be run to monitor the real time system performance.*

As indicated, our goal was to “*reflect real-world operating conditions*” (i.e., operating conditions where the mass spectrometer operated above minimum quality thresholds but still with varying sensitivity). For this reason, we could not specify a maintenance schedule at the outset, but instead we performed instrument maintenance when required according to QC data. We have modified the text of the revised manuscript to clarify for the reader that we did indeed use QC samples to monitor instrument performance. We defined “*as-needed*” by reference to these QC samples, as follows (underlined):

“To reflect real-world operating conditions, approximately 6,500 MS runs were carried out over a four-month period, during which the mass spectrometers received minor and major maintenance on an as-needed basis, indicated by quality control measures.”

-- Key Points

4.3 *We developed a novel strategy for MS data processing called ProNorM, which uses negative controls and technical replicates to normalise data from multiple instruments, reducing technical variation while strengthening biological signal.*

No response required.

4.4 *ProNorM rescues approximately 20% of values likely missing for technical reasons, by replacing these with plausible intensities from a distribution derived from replicates*

run on other instruments. The ProNorM method was developed to distinguish the real missing values due to low intensity (there is a concentration series in sample 1-8, there will be a trend. If the peptides have intensity values in high concentration, and NA in the low concert ration, they call it completely random NA) and missing values due to random reasons. ProNorM method seems to rely on the technical replicates. However, in the real DIA/SWATH analysis, 1) it is time consuming to run lots of duplicates for each sample, 2) the experimental intensities of peptide are unknown within different individual samples. Therefore, the ProNorM method seems not to work for real samples.

See Response 4.18.

4.5 DIA-MS data acquired over extended time periods from multiple instruments with differing maintenance schedules were reproducible, and changes in low-intensity peptide signals were detected in a dilution series comprising known ratios of human cancer tissues. I do not which data they used to conclude this. If it is Figure 2c, I will argue the “ovarian cancer-specific peptides” are mostly high abundant peptides in the ovarian tissue sample.

The data used to support this conclusion are shown in *Figures 3 and 5*.

Comments:

4.6 Figure 1a/b and Figure 2. The appearance order of yeast/ovarian/prostate in the names of each sample is different, making it a little hard to understand the results. Figure 1b. It is not very easy to understand the experiment design from Figure 1b, the numbers in 1a and 1b represent different meaning. Considering one 48h period serves as one independent cycle, a ring cycle may be a better way to show the workflow.

We have revised the ordering of panels in *Figure 1a* in the revised manuscript to be more consistent with the rest of the manuscript, as shown below.

Modifying *Figure 1b* to depict a cycle as suggested may give the impression that samples were run continuously throughout the four-month period, which was not the case. Instead, we have modified the figure to indicate the run order alongside the sample number in the revised manuscript. We have also modified the sample numbering in *Figure 1b* so that it is consistent with the numbering used in the rest of the manuscript, as shown below.

Figure 1. Study design. (a) Composition of the eight samples analysed repeatedly throughout the study.

Sample:

- S1 - 0% Ovary / 50% Prostate / 50% Yeast
- S2 - 3.125% Ovary / 50% Prostate / 46.875% Yeast
- S3 - 6.25% Ovary / 50% Prostate / 43.75% Yeast
- S4 - 12.5% Ovary / 50% Prostate / 37.5% Yeast
- S5 - 25% Ovary / 50% Prostate / 25% Yeast
- S6 - 50% Ovary / 50% Prostate / 0% Yeast
- S7 - 50% Ovary / 0% Prostate / 50% Yeast
- S8 - 100% HEK293T control cell line

Figure 1. Study design. ... (b) Twenty mass spectrometry runs during thirteen 48-hour periods on each of six instruments. Each run is represented by a coloured panel corresponding to each of the eight samples (labelled S1-S8), with the run order indicated in the upper left corner. Four samples were run in duplicate and four in triplicate during each 48-hour period.

4.7 Figure 2a. It is interesting to see there are some data points of sample 1-7 spreading out the grouping area in Figure 2a. How about the PCA analysis results after data normalization?

We provide *Revision Figure 10* below to show PCA plots using data after normalisation, coloured by sample (left) and instrument (right).

Revision Figure 10 – Principal component analysis of log₂-transformed experimental data after normalisation, with data points coloured by sample (left) and instrument (right). Missing values were filled with zero.

4.8 Page 6, line 154: Figure 2c. It is very important to show the intensity distribution of “frequently-observed human peptides” across sample 1-7, Figure 2e only show one example.

The purpose of *Figure 2* is to introduce the dataset to the reader, and to show some qualitative figures that demonstrate the dilution series in our data. In *Figure 2c*, we show data that are summarised from hundreds of peptides, in addition to the single peptide data presented in *Figures 2d* and *2e*. Most importantly, in *Figure 3* we show the correlation of $n = 2,904$ frequently-observed human peptides with the proportion of human tissue.

4.9 Page 6, line 158: The definition of “Ovarian cancer tissue-specific peptide” is unclear. According to my understanding, the definition of “Ovarian cancer tissue-specific peptide” might be (1) the “Ovarian cancer tissue-specific peptide” were filtered out based on the human peptides identified in both sample 1 and sample 7, all the other datasets from sample 2/3/4/5/6 were not used or (2) In summary, there were 156 runs of sample 1 and 7, respectively. Therefore, the appearance of these “Ovarian cancer tissue-specific peptide” should be < 8 runs in sample 1 (not containing ovarian tissue), and > 125 runs in sample 7 (not containing prostate

tissue). It will be much easier to understand the results if the authors describe the definition in details.

Ovarian cancer tissue-specific peptides are defined in the *Methods*:

“Ovarian cancer tissue-specific peptides (n = 849) were defined as peptides derived from human proteins that were present in more than 80% of runs containing ovarian cancer tissue but not prostate cancer tissue (Sample 7) and less than 5% of runs containing prostate cancer tissue but not ovarian cancer tissue (Sample 1).”

-- Methods, Data analysis and interpretation, Para. 3

4.10 Page 7, line 163: Figure 2d The numbers of samples containing quantification information of PSA peptides in sample 1 and 6 are much lower than sample 2-5 in Figure 2d. This is due to the different DIA-MS runs of sample 1/6 (n=156), and sample 2-5 (n=234). In order to make the data much clearer, it is better to show the relative proportion in each sample instead of absolute numbers in Figure 2d.

We have modified *Figures 2d* and *2e* in the revised manuscript, as suggested. The revised figures are shown below, with modified legend (see underlined text):

Figure 2. Baseline DIA-MS data reproducibility. ... (d, e) Intensities of all peptides identified from the (d) prostate-specific antigen (PSA) protein encoded by KLK3 and (e) the

housekeeping protein encoded by TARDBP. Boxplots show peptide intensity, with bar plots indicating the proportion of replicate samples in which each peptide was observed.

4.11 Page 7, line 166: Figure 2e. As the house-keeping protein TARDBP is constitutively expressed, the theoretical mass ratio of TARDBP in sample 1/2/3/4/5/6/7 should be 16:17:18:20:24:32:16. Based on the sample design, the TARDBP intensity in sample 1 and 7 should be the same. However, 1) the intensity of TARDBP in sample 7 is much higher than that of sample 1 in Figure 2e; 2) the number of samples containing TARDBP quantification information in sample 7 is also much higher than sample 1 in Figure 2e. How to explain these differences?

TARDBP is expressed in multiple human tissues. However, our data suggest that it is not expressed at the same intensity in both ovarian and prostate cancer tissues. This conclusion explains both observations noted, namely (a) the different intensity observed in Sample 1 and Sample 7 and (b) the different number of samples in which the peptides were quantified (more highly abundant peptides are generally quantified with fewer missing values). We have removed the term “*constitutively-expressed*” in the revised manuscript to avoid any confusion regarding the expected intensity of these peptides across tissue-types.

4.12 Page 7, line 172: Instrument sensitivity. 1) As shown in Figure 3a-b, the instrument sensitivity variation seems to be highly instrument specific. For example, “minor instrument clean” did not bring back the sensitivity of M02 and M06. However, the sensitivity gain of M05 after the “major instrument clean” is almost neglectful, the “minor instrument clean” already kept its sensitivity. Data from Figure 3a demonstrates that QC run is very useful to monitor the instrument sensitivity. As the authors run lot of other samples alongside this project, it is curious for them not to monitor the instrument sensitivity by QC in such big project. It is more meaningful to schedule the instrument maintenances based on QC results.

We were not surprised to find that maintenance had differing effects on the restoration of instrument performance. Where performance was restored after a minor instrument clean, we suggest that issues with the Q-jet or Q0 were the cause of the original performance degradation. In such circumstances, no further cleaning was required. Where maintenance of the Q-jet or Q0 was not sufficient to restore instrument performance, a major instrument clean was performed to target the Q1 or TOF entrance lens.

Instrument performance was indeed monitored by QC samples and metrics, and the results of running these QC samples indicated when instrument maintenance was required. For example, BSA protein is run every four samples for the purposes of mass calibration and to monitor MS/MS intensity. A sample of HEK293T is run every 16 samples to further monitor intensity. After instrument maintenance, instruments are checked using manufacturer provided standards to ensure that the instrument passes manufacturer specifications. To make this clearer to the reader, we now incorporate the following additional text into the revised manuscript (underlined):

“To reflect real-world operating conditions, approximately 6,500 MS runs were carried out over a four-month period, during which the mass spectrometers received minor and major maintenance on an as-needed basis, indicated by quality control measures.”

-- Key Points

4.13 The authors showed the MS2 level sensitivity data in Figure 3, it is very interesting to see the MS1 level sensitivity data. The reviewers sometimes found the sensitivity loss at MS2 level rather than MS1 level from the orbitrap type machine. How about the quantification dynamic range at both MS1 and MS2 level in this study?

In our laboratory, we monitor the total ion current (TIC) for each run of the mass spectrometer, along with monitoring of MS2 intensities from separate BSA and HEK293T digests, which are run for QC purposes. If we observe a drop in MS2 signal while the MS1 signal remains consistent, the instrument is cleaned to restore signal intensity. While DIA-MS does enable both MS1 and MS2 quantification, MS2 quantification is most commonly used. This topic has been reviewed in Pappireddi, Martin & Wühr *ChemBiochem* 2019 (see also Gillet et al *Mol Cell Proteomics* 2012, Rardin et al *Mol Cell Proteomics* 2015 and Collins et al *Mol Cell Proteomics* 2017). Further examination of this topic is out of the scope of this study.

4.14 Many researches claim that the sensitivity of DIA/SWATH is higher than DDA. In the original MS1 spectra, there will be both identified and non-identified peaks. It is interesting to show the precursor ions intensity proportion of each identified peptide in the total intensity of peaks in the corresponding MS1 spectra. This analysis may give a clear view of the dynamic range of DIA/SWATH.

We suggest that the key difference between DIA-MS and DDA-MS is not sensitivity, but rather is the lack of stochasticity. DIA-MS identifications will be limited by what spectral library is available for analysis. Even if a peak is present in the MS1, it will not be identified unless it is also present in the spectral library. Our purpose was to develop methods to ensure the reproducibility of the data obtained by large-scale DIA-MS, and not to identify all peaks that could be observed at the MS1 level. While we appreciate that exploring this question might be of interest, we feel that it is also beyond the scope of our study.

4.15 Page 174 and Figure 3b: The authors should clearly claim that the example human peptide was specifically selected from the “ovarian cancer tissue-specific peptides”.

As requested, we have modified the revised manuscript to specify this, as shown:

“To demonstrate the effects of long experimental periods on peptide intensities, a single human peptide that appeared to be ovarian cancer tissue-specific was examined in

detail.”

-- Results, Decrease in instrument sensitivity over time since cleaning, Para. 1

4.16 Page 8, line 206: Figure 3f: The theoretically mass of “frequently-observed human peptides” in sample 1-7 is different. Although the authors described that the intensity CV of each peptide was calculated for each sample in the figure legend, it is still not very clear to understand the data. It is better to show some example plots illustrating the intensity CV of each peptide from different samples.

We have modified *Figure 3f* in the revised manuscript to provide experiment-wide CVs per sample, rather than at different time points and instruments across the experiment, as shown:

Figure 3. Peptide intensity variation during the experimental period and normalisation approaches. ... (f) Coefficient of variation (CV) of frequently-observed human peptide intensities, calculated for each sample during the experimental period across all instruments. A black dashed line marks a CV of 15% for reference.

4.17 Page 11, line 291: Tissue proportion predication is pretty hard, as the real tissue-specific proteins and peptides should be used. In this study, the tissue proportion works well when the proportion difference of ovarian and prostate is very high, indicating the “Ovarian cancer tissue-specific peptide” dataset might be not very “tissue-specific”. As the number of ovarian tissue specific proteins may be very small, it is interesting for the authors to reduce the numbers of “Ovarian cancer tissue-specific peptide”, for example, only keep the peptides show very good linear intensity relationship to ovarian tissue proportion. What about the predication performance of this predication tool then? By the way, considering the yeast proteins will not be shared by ovarian and prostate tissue. How about the “tissue proportion” of yeast sample in sample 2-5?

Tissue-specific peptides were not used for the prediction of tissue proportions in *Figure 6*. We used the entire set of identified human peptides as input for the machine learning

algorithm. The model uses these input data along with the known proportion of ovarian cancer tissue for each sample in the training set to develop a model that can predict the proportion of ovarian cancer tissue in samples in the test set. We have modified the text to clarify this (underlined), as shown below. For more information, see Response 2.21.

*“To demonstrate whether such an analysis might be possible using DIA-MS data, we applied machine learning to estimate the proportion of ovarian cancer tissue among the prostate cancer tissue background after ProNorM. We trained four regularized regression models on the entire set of human peptides measured in three concentrations (for example, 6.25%, 12.5% and 25% ovarian cancer tissue; n = 695 samples) to avoid overfitting. Hyperparameters were tuned by ten-fold cross validation, and we report out-of-sample errors after testing the models on the remaining independent concentration (for example, 3.125% ovarian cancer tissue concentration; n = 225 samples; see **Methods; Fig. 6 and Supplementary Fig. 6**).”*

-- Results, Prediction of tissue proportion in a complex mixture, Para. 1

Regarding yeast proportions, these peptides are unsuitable for use after normalisation because we used yeast peptides for the purposes of normalisation.

4.18 Page 13, line 328: It is no doubt that increasing the number of technical replicates may be a useful strategy for analysis of small clinical cohorts. However, lots of replicates in DIA analysis lose its throughput advantage.

To provide more consideration of this point, we have modified the related text in the revised manuscript, as shown (underlined):

“Increasing technical replicate MS runs clearly comes at a cost. Increasing the number of technical replicates may be a useful strategy for analysis of small clinical cohorts, such as those available from rare diseases or clinical trials, or where researchers would derive benefit from a data matrix with fewer missing values. In the absence of technical triplicates for all samples, RUV-III-C can still be applied to a cohort independent of technical replacement, provided that at least some samples are replicated at various points during the experimental period.”

-- Discussion, Para. 3

Methods:

4.19 The DDA data were searched independently with three different search engines to generate the spectral library. Are there some MS/MS spectra that are matched to different peptides by the three search engines? If one MS/MS spectra was matched to different peptides, how to deal with this issue when combining the three results into one dataset?

These important questions have been addressed using published statistical frameworks and tools such as iProphet (Shteynberg et al *Mol Cell Proteomics* 2011) and PeptideShaker (Vaudel et al *Nat Biotechnol* 2015), and are widely used. In our study, the results from the

three search engines were merged using the PeptideShaker algorithm, as stated in the original manuscript. In brief, PeptideShaker converts the scores of the different search engines into Posterior Error Probability (PEP) values using a target/decoy strategy. Peptides mapping to both target and decoy are excluded. The product of search engine PEPs is given as a score for every peptide candidate. In cases where a MS/MS spectrum is matched to different peptides by multiple search engines, the best (i.e., the lowest scoring) peptide is selected. If two peptides are scored equally, then they are discriminated by:

- (i) the occurrence of their parent protein in the dataset;
- (ii) the number of search engines supporting the peptide;
- (iii) the number of fragment ions annotated in the spectrum; and
- (iv) the precursor mass error.

Further details regarding the PeptideShaker algorithm can be found in Vaudel et al *Nat Biotechnol* 2015.

Reviewers' comments:

Reviewer #1 (Remarks to the Author):

I think the authors addressed the comments by the reviewers. Thank you for addressing the reviews in this thorough fashion.

Reviewer #2 (Remarks to the Author):

While the authors have carefully responded to all the points raised in my original review, I am now even more sceptical about the suitability of the work for publication in Nature Communications. Rather than re-iterating the major points, the following three items may summarize my concerns.

a) the most original contribution is the software for data normalisation/missing value replacement. Yet, this software currently only works in the hands of the authors and it is unclear if it will be picked up by anyone producing more widely used data processing packages. Hence, the foreseeable impact beyond the contributing lab is unclear

b) The depth and quality of the data provided is actually quite low. Perhaps the choice of the Sciex hardware platform was not the right one but proteome coverage is not good by today's standards. This became clearer in the revision where the authors give details about how many peptides/proteins are in a sample. In their defense, the authors do report what they find and are not glossing over anything to make the data look better than it is. But the message that sticks to this reviewer is: "monitor your instrument performance regularly and keep it clean". This is entirely unsurprising and any large project would build in QC samples for this reason. It is not really clear how this manuscript "...enables large-scale proteomics...".

c) Perhaps the most disappointing part is that the manuscript implies that DIA is under-delivering on its many promises. The extent of missing data is actually quite high thus calling into question the perhaps most significant promise of DIA. If this is just down to the hardware/software used in this study, then the paper still does not make a strong point for DIA. Even though the authors do not agree, the data implies that sensitivity is also a real issue (in addition to data missing for software/peak detection reasons). Perhaps more important still, the data the authors present in terms of reproducible proteome coverage is in stark contrast to the figures reported by some of the leading DIA laboratories.

The authors clearly made the very laudable effort to analyse instrument related performance over quite a long time. In that sense, the work should be published. But this reviewer cannot see the impact commensurate with doing so in Nature Communications.

Point-by-point Response to Reviewers' Comments

Strategies to enable large-scale proteomics for reproducible research

Authors:

Rebecca C. Poulos^{1#}, Peter G. Hains^{1#}, Rohan Shah^{1#}, Natasha Lucas¹, Dylan Xavier¹, Srikanth S. Manda¹, Asim Anees¹, Jennifer M. S. Koh¹, Sadia Mahboob¹, Max Wittman¹, Steven G. Williams¹, Erin K. Sykes¹, Michael Hecker¹, Michael Dausmann¹, Merridee A. Wouters¹, Keith Ashman², Jean Yang³, Peter Wild⁴, Anna deFazio⁵, Rosemary L. Balleine¹, Brett Tully¹, Ruedi Aebersold⁶, Terence P. Speed⁷, Yansheng Liu⁸, Roger R. Reddel¹, Phillip J. Robinson¹, Qing Zhong^{1*}

Affiliations:

¹ ProCan[®], Children's Medical Research Institute, Faculty of Medicine and Health, The University of Sydney, Westmead, NSW, Australia.

² Sciex, 2 Gilda Court, Mulgrave, VIC, Australia.

³ School of Mathematics and Statistics, The University of Sydney, Sydney, Australia.

⁴ Dr. Senckenberg Institute of Pathology, University Hospital Frankfurt, Frankfurt am Main, Germany; and Department of Pathology and Molecular Pathology, University Hospital Zurich, Zurich, Switzerland.

⁵ Centre for Cancer Research, Westmead Institute for Medical Research, Westmead, NSW, Australia; Faculty of Medicine and Health, The University of Sydney, Westmead, NSW, Australia; and Department of Gynaecological Oncology, Westmead Hospital, Westmead, NSW, Australia.

⁶ Department of Biology, Institute of Molecular Systems Biology, ETH Zürich, Zürich, Switzerland; and Faculty of Science, University of Zürich, Zürich, Switzerland.

⁷ Bioinformatics Division, Walter and Eliza Hall Institute of Medical Research, Parkville, VIC, Australia; and Department of Mathematics and Statistics, University of Melbourne, Melbourne, VIC, Australia.

⁸ Department of Pharmacology, Yale University School of Medicine, New Haven, CT, USA; and Yale Cancer Biology Institute, Yale University, West Haven, CT, USA.

Notes:

These authors contributed equally; * Correspondence.

Reviewer #1

Comments:

1.1. I think the authors addressed the comments by the reviewers. Thank you for addressing the reviews in this thorough fashion.

No response required.

Reviewer #2

Comments:

While the authors have carefully responded to all the points raised in my original review, I am now even more sceptical about the suitability of the work for publication in Nature Communications. Rather than re-iterating the major points, the following three items may summarize my concerns.

2.1. The most original contribution is the software for data normalisation/missing value replacement. Yet, this software currently only works in the hands of the authors and it is unclear if it will be picked up by anyone producing more widely used data processing packages. Hence, the foreseeable impact beyond the contributing lab is unclear.

Our novel normalisation method is available alongside our manuscript as a standard R package, which will be released via CRAN upon publication. The usage instructions are clearly outlined in the package documentation. The RUV family of methods has widespread adoption in other 'omic fields and our research is vital in ensuring its utility for proteomic research. Furthermore, we demonstrate in our manuscript that our method can be successfully applied to independent datasets obtained from other laboratories, including *Collins et al 2017* (see **Fig S3e-f**) and *Bruderer et al 2019* (see **Supplementary Note**). The above comment implies (correctly) that our method is independent from other data processing pipelines. This is a strength of our work and will lead to more widespread adoption than if our method was confined to a single data processing pipeline used by only a subset of researchers. Analysing large-scale proteomic data of this kind routinely requires the integration of a number of standalone tools (see **Fig S1c**).

We also note that our research garnered significant interest at the Human Proteome (HUPO) World Congress 2019 and the Lorne Proteomics Symposium 2020, where we received repeated requests for access to our methods prior to publication. Our study has also been chosen for oral presentation at the American Society for Mass Spectrometry (ASMS) Conference 2020. By all available indications, our novel tool will be of great interest to the scientific community.

We have modified the revised manuscript to highlight the availability of our R package via CRAN, as follows:

*“The RUV-III-C implementation is available as an R package in **Supplementary File 2**, which will be stored at the Comprehensive R Archive Network (CRAN) upon publication.”*

-- Methods, Data and code availability, Para. 1

2.2. The depth and quality of the data provided is actually quite low. Perhaps the choice of the Sciex hardware platform was not the right one but proteome coverage is not good by today's standards. This became clearer in the revision where the authors give details about how many peptides/proteins are in a sample. In their defense, the authors do report what they find and are not glossing over anything to make the data look better than it is. But the message that sticks to this reviewer is: "monitor your instrument performance regularly and keep it clean". This is entirely unsurprising and any large project would build in QC samples for this reason. It is not really clear how this manuscript "...enables large-scale proteomics..."

We did not undertake a study aimed at showing that DIA (or Sciex) measurements are reproducible across instruments in terms of the greatest possible numbers of protein identifications. This has been attempted by others. Our purpose was to develop approaches to improve the reproducibility of large-scale, high-throughput DIA-MS measurements, thus enabling quantitative accuracy across instruments and over extended periods of time. We intentionally undertook this in the context of realistic operating conditions in a large cancer facility. We assessed reproducibility by analysing complex human cancer tissue samples, rather than using ideal fresh tissue or synthetic dilutions of simple peptide mixtures. Thus, there were trade-offs to achieve our objective. We required a large amount of peptides to obtain sufficient material for thousands of identical runs. We therefore utilised exceptionally large tumour samples comprising connective tissue, necrotic regions, even ink and other contaminants. The starting samples were not carefully dissected for optimum tumour content, nor were the highest peptide loads used. We do not attempt to demonstrate how our instruments might perform at protein depth of coverage, and it is therefore unsurprising that we detected fewer proteins than observed in other cohorts. This does not reflect poor data quality, and we do not attempt to make a statement regarding the technology or choice of MS vendor. We routinely identify more than 7,500 proteins in our spectral libraries and quantify more than 5,000 proteins by SWATH-MS when analysing higher quality micro-dissected samples or cell lines under standard conditions in our laboratory, with identical instrument loadings. Therefore, the depth of our data here reflects the actual samples analysed and is not relevant to the message of our manuscript. To make this point more explicit in the revised manuscript, we have modified the following text (underlined):

“This study of data collected from 1,560 DIA-MS runs of complex samples, in a single facility over four months, provided an unprecedented opportunity to develop new

approaches that enable large-scale tissue proteomic data to be mined for improved reproducibility over time and between instruments. Tissue samples were selected to ensure that a sufficient amount of peptide was available to complete the large number of MS runs, and the experiment was not optimised for an in-depth analysis of the prostate or ovarian cancer proteome.”

-- Discussion, Para 1.

It is not correct that the main message of our manuscript is to “monitor your instrument performance regularly and keep it clean”. We required the instruments to operate in a real-world scenario that included the need for cleaning. This was an experimental parameter, rather than a goal that we were aiming to improve. Our instruments operated in a high-quality setting and well above minimum QC levels. In the proteomics field, it is simply axiomatic that instrument performance degrades with time after cleaning. MS performance needs to be balanced with the cleaning schedule, as it is impractical to run such a large number of samples when constant cleaning is required. This represents a major barrier to the generation of large datasets, which is a critical point that currently limits the advancement of the whole proteomics field. The significance that has been overlooked is that our new computational approach shows how to overcome this barrier to enable accurate and reproducible large-scale proteomics, regardless of time and instrument cleaning schedules. We have modified the revised manuscript to improve the clarity of our message. Please refer to the marked manuscript for details of these specific changes.

2.3. Perhaps the most disappointing part is that the manuscript implies that DIA is under-delivering on its many promises. The extent of missing data is actually quite high thus calling into question the perhaps most significant promise of DIA. If this is just down to the hardware/software used in this study, then the paper still does not make a strong point for DIA. Even though the authors do not agree, the data implies that sensitivity is also a real issue (in addition to data missing for software/peak detection reasons). Perhaps more important still, the data the authors present in terms of reproducible proteome coverage is in stark contrast to the figures reported by some of the leading DIA laboratories. The authors clearly made the very laudable effort to analyse instrument related performance over quite a long time. In that sense, the work should be published. But this reviewer cannot see the impact commensurate with doing so in Nature Communications.

We do not believe that it is a valid concern that the publication of a study that has relatively low proteome coverage will turn people away from DIA as a technology. We fully accept that other carefully micro-dissected samples can and do yield higher numbers of protein IDs. We have addressed this in Response 2.2 above, and have modified the revised manuscript accordingly. Further, we do not attempt to make quantitative statements about what proportion of missing values might be acceptable in a dataset (this has been addressed by others; see Bruderer et al 2017 and Selevsek et al 2015). Our study significance is that we provide a method for reducing the extent of data missingness by effectively utilising technical

replicates, in a manner not previously envisaged. The above comment overlooks the purpose of the paper, which is that we have developed a method to overcome a key limitation of mass spectrometers, to enable, for the first time, the normalisation and quantitation of large-scale, reproducible proteomic datasets, using multiple instruments across considerable periods of time and with disparate cleaning states. It is currently not possible to effectively do this using existing approaches. We therefore anticipate that our approach will become widely used, and thus we consider that this manuscript is highly suitable for publication in *Nature Communications*.